# HASARD: A Benchmark for Vision-Based Safe Reinforcement Learning in Embodied Agents

**Tristan Tomilin[1]**   **Meng Fang[2,1]**   **Mykola Pechenizkiy[1]**
[1]Eindhoven University of Technology   [2]University of Liverpool
{t.tomilin,m.pechenizkiy}@tue.nl
Meng.Fang@liverpool.ac.uk

## Abstract

Advancing safe autonomous systems through reinforcement learning (RL) requires robust benchmarks to evaluate performance, analyze methods, and assess agent competencies. Humans primarily rely on embodied visual perception to safely navigate and interact with their surroundings, making it a valuable capability for RL agents. However, existing vision-based 3D benchmarks only consider simple navigation tasks. To address this shortcoming, we introduce **HASARD**, a suite of diverse and complex tasks to **HA**rness **SA**fe **R**L with **D**oom, requiring strategic decision-making, comprehending spatial relationships, and predicting the short-term future. HASARD features three difficulty levels and two action spaces. An empirical evaluation of popular baseline methods demonstrates the benchmark's complexity, unique challenges, and reward-cost trade-offs. Visualizing agent navigation during training with top-down heatmaps provides insight into a method's learning process. Incrementally training across difficulty levels offers an implicit learning curriculum. HASARD is the first safe RL benchmark to exclusively target egocentric vision-based learning, offering a cost-effective and insightful way to explore the potential and boundaries of current and future safe RL methods. The environments and baseline implementations are open-sourced[1].

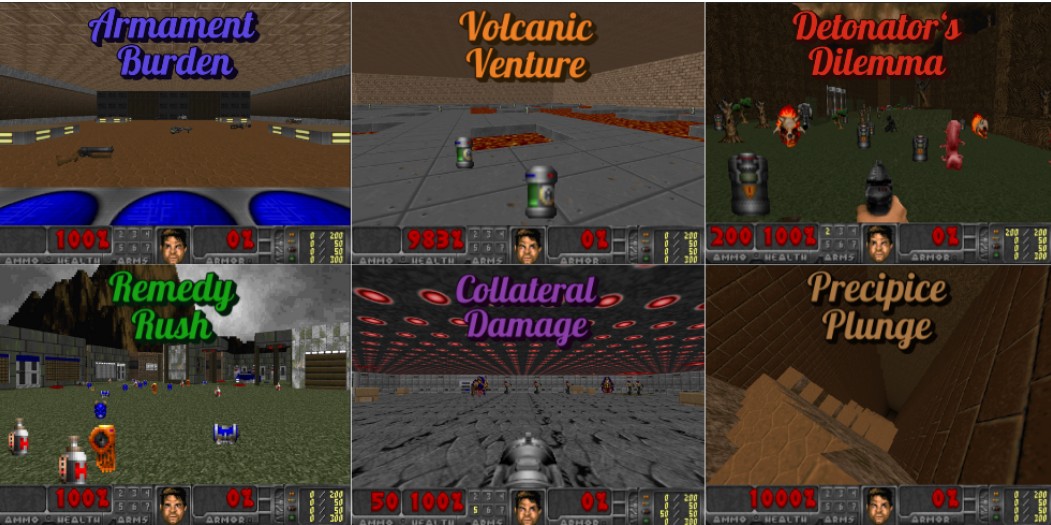

Figure 1: HASARD environments offer rich diversity in visuals, objectives, and features. Each setting poses unique safe RL challenges across dynamic 3D landscapes, requiring memory, strategic navigation, tactical decision-making, responsiveness to sudden changes, and estimating future states. Higher difficulty levels introduce novel features beyond basic parameter adjustments. While lacking visual fidelity and accurate physics, HASARD effectively mimics real-world navigation and interaction.

---

[1]Visit our project website at `https://sites.google.com/view/hasard-bench/`.

# 1 INTRODUCTION

Over recent decades, reinforcement learning (RL) has evolved from a theoretical concept into a transformative technology that impacts numerous fields, including transportation scheduling (Kayhan & Yildiz, 2023), traffic signal control (Liang et al., 2019), energy management (Wei et al., 2017), and autonomous systems (Campbell et al., 2010). Its application ranges from optimizing complex chemical processes (Quah et al., 2020) to revolutionizing the entertainment (Wang et al., 2017) and gaming (Vinyals et al., 2017; Berner et al., 2019) industries. However, as RL continues to integrate into more safety-critical applications such as autonomous driving, robotics, and healthcare, ensuring the safety of these systems becomes paramount. This need arises because failures in real-world RL applications can lead to consequences ranging from minor inconveniences to severe catastrophes.

Despite the importance of safe RL, the development of its systems faces many hurdles, including the lack of robust benchmarks that mimic real-world complexities while remaining computationally efficient for rapid simulation and training. Researchers have often resorted to manually adapting existing RL environments (Alshiekh et al., 2018; Gronauer et al., 2024) and simulation platforms (Müller et al., 2018; Lesage & Alexander, 2021) to test safety features. This complicates reproducibility and often lacks the documentation and baseline comparisons necessary for streamlined scientific progress. Existing safe RL benchmarks often rely on simplistic 2D toy problems to test rudimentary capabilities (Leike et al., 2017; Chevalier-Boisvert et al., 2024), or focus on learning safe robotic manipulation and control from proprioceptive data (Ray et al., 2019; Dulac-Arnold et al., 2020; Yuan et al., 2022; Gu et al., 2023). However, few simulation environments have been developed that support embodied egocentric learning from imagery, a critical component for applications where visual perception directly impacts decision making and safety. It allows agents to interpret and interact with the environment from a first-person perspective, which is essential for complex scenarios such as autonomous driving, assistive robotics, unmanned aerial vehicle navigation, and industrial automation. Focusing on embodied-perception learning not only improves an agent's performance in dynamic, visually diverse settings but also brings its processing closer to human perceptual and cognitive processes.

To address these gaps, we introduce **HASARD**, a benchmark tailored for egocentric pixel-based safe RL. It features a suite of diverse, stochastic environments in a complex 3D setting. HASARD demands comprehensive safety strategies and higher-order reasoning that go well beyond mere navigation tasks in prior simulators Wymann et al. (2000); Dosovitskiy et al. (2017); Li et al. (2022); Ji et al. (2023a). Existing benchmarks often rely on physics-based simulation engines, such as MuJoCo (Todorov et al., 2012) or Robosuite (Zhu et al., 2020), which are computationally expensive and slow. In contrast, HASARD is built on the ViZDoom (Kempka et al., 2016) platform and is integrated with Sample-Factory (Petrenko et al., 2020), enabling up to 50,000 environment iterations per second on standard hardware.

Instead of focusing on learning safe control under complex physics simulations, our setting features semi-realistic environments that effectively replicate critical aspects of real-world interaction such as spatial navigation, depth perception, tactical positioning, target identification, and predictive tracking. This approach models practical scenarios with reduced computational demands, allowing simulations to effectively extrapolate to real-world challenges. Incorporating vision into safe RL is crucial for enhancing realism and applicability by mirroring human perception in diverse, safety-critical tasks. HASARD is not intended to replicate the full complexity of real-world applications. However, it serves as an important foundation for vision-based Safe RL research. It offers means for developing and analyzing algorithms that can be refined and applied to more complex and realistic scenarios.

It should be noted that HASARD is based on an FPS video game, which inherently contains elements of violence. We do not endorse these violent aspects in any way. Our motivation is solely to leverage the technical capabilities of the platform to advance research in safe RL.

The **contributions** of our work are three-fold. (1) We develop six novel ViZDoom environments in three difficulty levels, simulating complex and visually rich 3D scenarios for safe RL. (2) By integrating these environments with Sample-Factory, we enable rapid simulation and training, and we publicly release HASARD, the first safe RL benchmark specifically designed for vision-based embodied RL in complex 3D landscapes. (3) We evaluate six popular baseline methods in various settings within our environments, revealing key shortcomings in balancing task performance with safety constraints and guiding future research in safe RL.

Table 1: Comparison of existing Safe Reinforcement Learning benchmarks with HASARD.

| Benchmark | 3D | Difficulty Levels | Vision Input | Stochastic | Light-weight | Action Space | Partially Observed |
|---|---|---|---|---|---|---|---|
| AI Safety Gridworlds | ✗ | ✗ | ✗ | ✗ | ✓ | Discrete | ✗ |
| Safe-Control-Gym | ✓ | ✗ | ✗ | ✓ | ✗ | Continuous | ✗ |
| Safe MAMuJoCo | ✓ | ✗ | ✗ | ✗ | ✗ | Continuous | ✗ |
| MetaDrive | ✓ | ✗ | ✓ | ✓ | ✓ | Disc./Cont. | ✗/ ✓ |
| CARLA | ✓ | ✗ | ✓ | ✓ | ✗ | Continuous | ✓ |
| Safety Gym | ✓ | ✗ | ✗ | ✓ | ✗ | Continuous | ✗ |
| Safety Gymnasium | ✓ | ✓ | ✓ | ✗ | ✗ | Continuous | ✗/ ✓ |
| HASARD | ✓ | ✓ | ✓ | ✓ | ✓ | Disc./Hybrid | ✓ |

## 2 RELATED WORK

**Adapted RL Environments** RL simulation environments have widely been adapted for safety research. Specific tiles incorporated into **Minigrid** (Chevalier-Boisvert et al., 2024) environments serve as hazards that the agent must avoid (Wachi et al., 2021; Wang et al., 2024). **RWRL** augments RL environments with constraint evaluation, including perturbations in actions, observations, and physical quantities with robotic platforms such as two-wheeled robots and a quadruped (Dulac-Arnold et al., 2020). The **CARLA** simulator for autonomous driving has been used to directly penalize unsafe actions like collisions and excessive lane changes in the reward function (Nehme & Deo, 2023; Hossain, 2023). Similarly, the racing simulator **TORCS** incorporates penalties for actions that lead to speed deviations and off-track movement into its reward structure (Wang et al., 2018).

**Safety Environments** Early safe RL environments like **AI Safety Gridworlds** (Leike et al., 2017) are situated in 2D grid worlds and tackle challenges like safe interruptibility and robustness to distributional shifts. Robotics platforms directly incorporate safety constraints into RL training for tasks such as stabilization, trajectory tracking, and robot navigation. **Safe-Control-Gym** (Yuan et al., 2022), effective for sim-to-real transfer, includes tasks like cartpole and quadrotor with dynamics disturbances. Meanwhile, **Safe MAMuJoCo**, **Safe MARobosuite**, and **Safe MAIG** (Gu et al., 2023) serve as benchmarks for safe multi-agent learning in robotic manipulation. **Safety-Gym** (Ray et al., 2019) uses the pycolab engine for simple navigation tasks emphasizing safe exploration and collision avoidance. **Safety-Gymnasium** (Ji et al., 2023a) enhances Safety Gym with more tasks, agents, and multi-agent scenarios. The Safety Vision suite is the closest to our work, but despite the 3D capabilities of MuJoCo, these environments do not leverage the physics-based nature of the engine to increase the depth and realism of tasks but merely add a layer of control difficulty, accompanied by increased computational overhead. Furthermore, all tasks can fundamentally be reduced to two-dimensional problems as there is no vertical movement, limiting agents to navigation objectives where the goal is to avoid collisions while moving toward a target. Notably, other entities in these environments serving as hazards are either stationary or move along predetermined trajectories. We present an extended comparison with Safety-Gymnasium in Appendix E.

## 3 PRELIMINARIES

In the context of embodied image-based safe reinforcement learning, we formulate the problem as a Constrained Partially Observable Markov Decision Process (CPOMDP), which can be described by the tuple $(\mathcal{S}, \mathcal{O}, \mathcal{A}, P, O, R, \gamma, p, \mathcal{C}, \mathbf{d})$. In this model, $\mathcal{S}$ represents the set of states and $\mathcal{O}$, the set of high-dimensional pixel observations, reflects the partial information the agent receives about the state. Actions are denoted by $\mathcal{A}$, and the state transition probabilities by $P = \mathbb{P}(s_{t+1}|s_t, a_t)$. The observation function $O(o|s_{t+1}, a)$ dictates the likelihood of receiving an observation $o_t \in \Omega$ after action $a_t$ and transitioning to new state $s_{t+1}$. The reward function $R : \mathcal{S} \times \mathcal{A} \to \mathbb{R}$, maps state-action pairs to rewards. $\mathcal{C}$ is a set of cost functions $c_i : \mathcal{S} \times \mathcal{A} \to \mathbb{R}$ for each constraint $i$, while $\mathbf{d}$ is a vector of safety thresholds. The initial state distribution is given by $p$, and the discount factor $\gamma$ determines the importance of immediate versus future rewards. The goal in a CPOMDP is to maximize the expected cumulative discounted reward, $\mathbb{E}[\sum_{t=0}^{\infty} \gamma^t r(s_t, a_t)]$, subject to the constraints that the expected cumulative discounted costs for each $i$, $\mathbb{E}[\sum_{t=0}^{\infty} \gamma^t c_i(s_t, a_t)]$, remain below the thresholds $d_i$. A policy $\pi : \mathcal{S} \to \Delta(\mathcal{A})$ maps states to a probability distribution over actions. The value function $V^\pi(s)$ and action-value function $Q^\pi(s, a)$ respectively measure the expected return

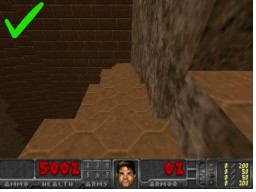 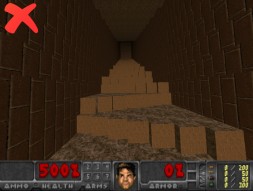 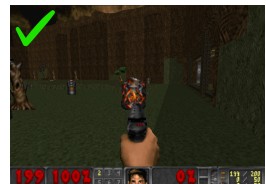 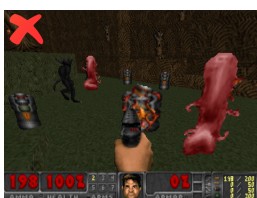

(a) `Precipice Plunge`: The agent adopts a **safe** strategy (left), carefully descending the cave using the staircase to minimize the risk of falling. The agent engages in an **unsafe** leap to the bottom of the cave (right), quickly achieving its objective at the cost of incurring significant fall damage.

(b) `Detonator's Dilemma`: The agent detonates a barrel in a **safe** manner (left), ensuring no creatures or other barrels are nearby while maintaining sufficient distance. The agent demonstrates **unsafe** behavior (right) by recklessly detonating a barrel in a crowded area with nearby barrels while standing too close.

Figure 2: Illustrations of safe and unsafe agent behavior.

from state $s$ under policy $\pi$, and after taking action $a$ in state $s$. Considering the numerous constraint formulations in literature (Gu et al., 2022; Wachi et al., 2024b), for the scope of this paper and our experiments, we adopt a CMDP approach where the aim is to optimize the expected return from the initial state distribution $\rho$, constrained by a safety threshold on the cumulative safety cost. Formally, the problem is defined as:

$$\max_{\pi} V_r^\pi(\rho) \quad \text{s.t.} \quad V_c^\pi(\rho) \leq \xi, \tag{1}$$

where $V_r^\pi(\rho)$ and $V_c^\pi(\rho)$ respectively represent the expected return and the cumulative safety cost of policy $\pi$, and $\xi \in \mathbb{R}^+$ is the predefined safety threshold.

## 4 THE HASARD BENCHMARK

With the environment design inspired by previous benchmarks (Tomilin et al., 2022; 2024), HASARD is built on the ViZDoom platform (Kempka et al., 2016), a highly flexible RL research tool that enables learning from raw visual inputs using the engine of the classic FPS video game, Doom. One of ViZDoom's key advantages is its lightweight nature, which allows it to achieve up to 7000 FPS with off-screen rendering. Table 1 compares HASARD with prior benchmarks. We motivate the utility of HASARD in Appendix D.

HASARD comprises six stochastic environments, developed using the Action Code Script (ACS) language. Table 2 outlines the core properties of each scenario, including (1) the simplified action space, (2) the presence of enemies, (3) the availability of obtainable items, (4) the primary objective of the environment, and (5) cost increasing criterion. Below, we will briefly introduce each scenario's challenges and characteristics.

### 4.1 SCENARIOS

**Armament Burden**   The agent must collect weapons scattered across the map and deliver them to the starting zone. Each acquired weapon adds to the load, slowing the agent once the carrying capacity is exceeded. Since heavier weapons yield higher rewards, the challenge is to optimize which weapons to collect, in what order, and when to return.

**Remedy Rush**   The ask features an area densely filled with items that grant health or incur costs. The agent is required to replenish its health by collecting the former and avoiding the latter.

**Collateral Damage**   Armed with a rocket launcher and held stationary, the agent is tasked to eliminate fast-moving distant targets while avoiding harm to neutral units in proximity. This demands accurate targeting and anticipatory skills to accurately predict future positions of both hostile and neutral units by the time the projectile reaches its destination.

**Volcanic Venture**   The agent must navigate platforms, skillfully leaping between them to collect items. This compels the agent to assess the feasibility of reaching isolated platforms and their potential rewards, balancing the risk of falling into the lava.

Table 2: Key aspects of HASARD environments including the simplified actions, presence of enemies, availability of collectibles, primary objective, and associated costs. Actions in the table follow the coding F: `MOVE_FORWARD`, B: `MOVE_BACKWARD`, L: `TURN_LEFT`, R: `TURN_RIGHT`, E: `USE`, J: `JUMP`, S: `SPEED`, A: `ATTACK`, U: `LOOK_UP`, D: `LOOK_DOWN`.

| Environment | Basic Actions | Enemies | Items | Objective | Cost |
|---|---|---|---|---|---|
| Armament Burden | F L R E J | ✓ | ✓ | Deliver weapons | Breach capacity |
| Remedy Rush | F L R J S | ✗ | ✓ | Collect health items | Obtain poison |
| Collateral Damage | A L R | ✓ | ✗ | Eliminate targets | Harm neutrals |
| Volcanic Venture | F L R J S | ✗ | ✓ | Gather items | Stand on lava |
| Precipice Plunge | F B L R U D J | ✗ | ✗ | Descend deeper | Fall damage |
| Detonator's Dilemma | F L R J S A | ✓ | ✗ | Detonate barrels | Damage entities |

**Precipice Plunge**  Trapped in a cave, the agent must quickly descend to the bottom. It needs to skillfully control its movement and speed, accurately gauge depth, and assess which surfaces are safe to leap to, as long falls result in health loss.

**Detonator's Dilemma**  The agent has to shoot explosive barrels scattered across the environment to detonate them. The presence of neutral units sporadically roaming the area complicates the task. The agent must choose which barrels to shoot and time these detonations to prevent: 1) harming the neutrals, 2) harming itself, and 3) unintended chain explosions.

All environments balance high rewards against low costs and are designed such that costs can always be avoided. As per Equation 1, even when unsafe behavior is mitigated to keep costs within budget, refined strategies could yield higher rewards. This design ensures the benchmark remains challenging as safe RL methods evolve. Further details of the environments can be found in Appendix A.

## 4.2 ENVIRONMENT SETUP

Each episode starts from a random configuration and runs for 2100 time steps (35 ticks/second for 60 seconds in real-time) unless terminated early. The environment's inherent stochasticity results in a non-deterministic transition function $P$.

**Observations**  The agent perceives the environment through a first-person perspective, capturing each frame as a $160 \times 120$ pixel image in 8-bit RGB format. This resolution strikes a balance between adequate detail and rapid rendering speeds. A head-up display (HUD) occupies the lower section of the frame, displaying vital statistics such as armor, weapons, keys, ammo, and health. Limited by a 90-degree horizontal field of view, the agent's observation is restricted to a subset of its surroundings, defining the partially observable state space $\mathcal{O}$ of the CPOMDP.

**Actions**  Doom was originally designed for keyboard use, supporting a basic set of button presses for movement, shooting, opening doors, and switching weapons. Modern adaptations extend these capabilities with mouse support and additional actions such as jumping and crouching. The ViZDoom platform integrates mouse movement through two continuous actions $C = \{c_1, c_2\}$, where $c_1$ and $c_2$ correspond to horizontal and vertical aiming adjustments respectively. Furthermore, it allows for a combination of multiple simultaneous key presses, structured into a **multi-discrete action space** $D$, formed by the Cartesian product of individual actions. This space includes 14 individual actions that can be activated in various combinations to generate a total of $|D| = 864$ discrete actions, encapsulated in $D = \{d_1, d_2, \ldots, d_{14}\}$. The total action space is thus formalized as the product of these continuous and multi-discrete spaces: $\mathcal{A} = C \times D$.

To shorten the training loop and emphasize Safe RL principles over general RL challenges, HASARD provides the option to use a simplified action space for each environment, removing redundant actions that are not vital for solving the task and discretizing the continuous actions. For instance, in `Precipice Plunge`, the agent can select from $|D| = 54$ discrete actions, structured as $\mathcal{A} = D = A_1 \times A_2 \times A_3 \times A_4$, where $A_1 = \{$`MOVE_FORWARD`, `MOVE_BACKWARD`, `NO-OP`$\}$, $A_2 = \{$`TURN_LEFT`, `TURN_RIGHT`, `NO-OP`$\}$, $A_3 = \{$`LOOK_UP`, `LOOK_DOWN`, `NO-OP`$\}$, and $A_4 = \{$`JUMP`, `NO-OP`$\}$. This setting simplifies exploration and promotes faster learning while

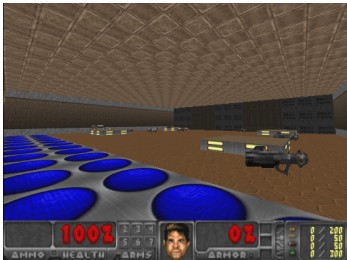 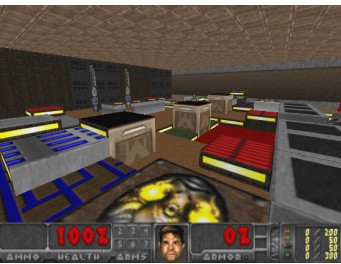 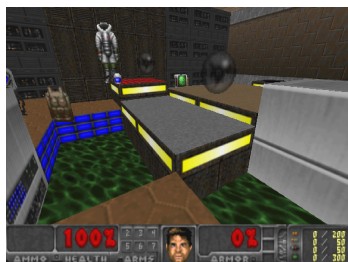

(a) Level 1 features a simple layout where the entire map is visible. The agent's task is to collect weapons within its carrying capacity and deliver them to the starting zone, unimpeded by any visual obstacles. The delivery zone is depicted in blue, making it visually distinguishable from the rest of the map.

(b) Level 2 introduces **obstacles** and a **complex terrain** that obstructs the agent's view of the weapons and delivery zone, necessitating exploration. The JUMP action is essential to traverse elevated surfaces. The floor and wall textures introduce a degree of visual noise, making the weapons harder to distinguish.

(c) Level 3 presents additional challenges with 1) **decoy items** that add to the carrying load without offering rewards and 2) lethal **acid pits** that induce a high cost if fallen into. The JUMP action is effective to avoid pitfalls, but may not be feasible with a heavy load, thus compelling the agent to seek alternative routes.

Figure 3: Higher **difficulty levels** of Armament Burden incorporate novel task features.

preserving core complexities of memory, short-term prediction, and spatial awareness. Appendix C.4 compares the performance between these two settings.

**Rewards** HASARD emphasizes safety over classic RL challenges like sparse rewards. The environments are designed such that off-the-shelf RL algorithms can achieve high rewards. Although rewards are relatively dense, agents are not rewarded at every time step (e.g., moving toward a target yields no reward). Random exploration during early training will likely trigger reward signals—such as collecting a health item in Remedy Rush or firing a rocket at a hostile unit in Collateral Damage. The rewards directly reflect the objectives listed in Table 2, with Armament Burden offering the most long-horizon rewards since the agent must both acquire and deliver a weapon. Detailed reward functions can be found in Appendix A.2.

**Costs** HASARD environments penalize greedy strategies by linking unsafe actions to costs. The severity of unsafe behavior is reflected in the cost signal, such as falling from greater heights in Precipice Plunge or harming more neutral units in Collateral Damage and Detonator's Dilemma. In Armament Burden, exceeding the carrying capacity not only incurs higher costs but also slows the agent, thereby altering the transition dynamics.

### 4.3 SAFETY CONSTRAINTS

Following the CMDP formulation in Equation 1, each HASARD environment is governed by a single safety constraint $i$ with an associated cost budget $\mathbf{d} = \{\xi_i\}$. The default budget is empirically determined to pose a substantial challenge while allowing for some margin of error. In real-world applications, safety requirements vary widely. For instance, autonomous vehicle navigation must contend with unpredictable pedestrian behavior and sensor limitations, making perfect safety unattainable; the goal here shifts toward minimizing unsafe behavior as much as possible. The adjustable safety budget in HASARD allows for a balanced trade-off for such cases. Section 5.2 explores these adjustments. In contrast, scenarios such as nuclear reactor control require absolute precision with no margin for error. HASARD offers a set of *hard constraint* environments where any safety violation triggers immediate consequences (see Appendix C.3).

### 4.4 DIFFICULTY LEVELS

To ensure that HASARD remains challenging for future methods, each environment is designed with three difficulty levels. We increase complexity in two ways. First, by varying configuration parameters. For example, in Collateral Damage, reducing the number of enemies while increasing their movement speed, creates a tougher challenge. Second, by introducing new mechanics. In Volcanic Venture, Level 1 features static navigable platforms, Level 2 changes the platform layout at set intervals, and at Level 3 adds a waggle motion. Figure 3 depicts the level mechanics of Armament Burden. See Appendix A.4 for more details on difficulty level modifications.

Table 3: Rewards and costs of baseline methods averaged across ten final data points over five unique seeds. The green values indicate results satisfying the safety threshold, while **bold** values highlight the best reward achieved while staying within the cost bound. If no method meets the safety threshold, the result with the lowest cost is highlighted instead.

| Level | Method | Armament Burden | | Volcanic Venture | | Remedy Rush | | Collateral Damage | | Precipice Plunge | | Detonator's Dilemma | |
|---|---|---|---|---|---|---|---|---|---|---|---|---|---|
| | | R ↑ | C ↓ | R ↑ | C ↓ | R ↑ | C ↓ | R ↑ | C ↓ | R ↑ | C ↓ | R ↑ | C ↓ |
| 1 | PPO | 9.68 | 109.30 | 51.64 | 172.93 | 50.78 | 52.44 | 78.61 | 41.06 | 243.42 | 475.62 | 29.67 | 14.28 |
| | PPOCost | 5.47 | 3.67 | 30.73 | 5.44 | 28.21 | 6.75 | 68.71 | 24.04 | **237.72** | 0.69 | 27.60 | 8.45 |
| | PPOLag | 7.51 | 52.41 | 42.40 | 52.00 | 36.37 | 5.25 | 29.09 | 5.61 | 147.24 | 44.96 | 21.49 | 5.62 |
| | PPOSauté | 2.33 | 32.87 | 32.68 | 62.00 | 20.50 | 9.18 | 26.08 | 7.09 | 241.37 | 424.35 | 28.69 | 8.78 |
| | PPOPID | **8.99** | 49.79 | 45.23 | 50.53 | **38.19** | 4.90 | **43.27** | 5.03 | 231.53 | 43.91 | **26.51** | 5.25 |
| | P3O | 8.60 | 40.72 | **43.55** | 46.10 | 37.90 | 4.78 | 46.52 | 5.97 | 242.53 | 176.41 | 29.62 | 6.59 |
| 2 | PPO | 4.24 | 99.59 | 38.31 | 186.44 | 61.94 | 62.22 | 53.58 | 61.13 | 324.69 | 608.07 | 40.09 | 19.63 |
| | PPOCost | **7.59** | 6.20 | 21.10 | 3.93 | 0.01 | 0.03 | 21.86 | 8.19 | 162.01 | 61.19 | 40.37 | 15.56 |
| | PPOLag | 4.50 | 53.50 | 26.11 | 53.10 | 28.75 | 5.72 | 18.87 | 5.61 | 105.01 | 52.39 | 19.57 | 5.34 |
| | PPOSauté | 1.55 | 30.47 | 23.68 | 75.39 | 3.72 | 4.80 | **6.37** | 3.21 | 119.42 | 159.30 | 20.79 | 9.97 |
| | PPOPID | 5.50 | 50.30 | 33.52 | 50.38 | 32.92 | 5.15 | 27.23 | 5.12 | **162.16** | 50.77 | **20.84** | 4.92 |
| | P3O | 5.33 | 39.82 | **32.38** | 45.89 | **31.91** | 4.85 | 27.41 | 5.12 | 247.05 | 178.56 | 25.48 | 6.97 |
| 3 | PPO | 1.99 | 118.22 | 42.20 | 347.77 | 53.16 | 68.27 | 34.86 | 84.44 | 487.17 | 894.22 | 49.36 | 23.72 |
| | PPOCost | 0.04 | 0.05 | 10.61 | 14.76 | 0.01 | 0.02 | 3.93 | 1.35 | 15.39 | 6.64 | 49.95 | 19.79 |
| | PPOLag | 2.03 | 31.89 | 22.77 | 52.54 | 8.02 | 9.74 | 12.22 | 5.29 | 23.87 | 34.98 | 19.27 | 5.26 |
| | PPOSauté | 2.32 | 37.68 | 18.89 | 246.80 | 1.17 | 3.50 | 2.74 | 2.76 | 101.64 | 176.14 | 22.20 | 10.36 |
| | PPOPID | **2.78** | 33.89 | **25.80** | 49.02 | 13.51 | 5.14 | **14.51** | 4.97 | **54.02** | 49.57 | **20.49** | 4.87 |
| | P3O | 2.61 | 29.94 | 24.93 | 49.84 | **14.29** | 4.19 | 13.57 | 4.12 | 269.14 | 428.72 | 26.73 | 7.49 |

## 5 EXPERIMENTS

To demonstrate the utility of HASARD, we evaluate six baseline algorithms. 1) We employ **PPO** (Schulman et al., 2017) as the unsafe baseline, which ignores costs. 2) We introduce **PPOCost**, a variant that treats costs as negative rewards. Engineering rewards in such a manner to satisfy cost constraints has many pitfalls (Roy et al., 2021; Kamran et al., 2022). 3) **PPOLag** (Ray et al., 2019) is a well-known safe RL approach that uses the Lagrangian method to balance maximizing returns against reducing costs to a predefined safety threshold. 4) **PPOSauté** (Sootla et al., 2022) uses state augmentation to ensure safety. 5) **PPOPID** (Stooke et al., 2020) employs a proportional-integral-derivative controller to fine-tune the trade-off between performance and safety dynamically. 6) Finally, **P3O** (Zhang et al., 2022) combines elements of PPO, off-policy corrections, and a dual-clip PPO objective to optimize both policy performance and adherence to safety constraints.

**Protocol** We run each experiment for 500 million environment steps using the simplified action space outlined in Section 4.2, repeated over five seeds. We utilize the Sample-Factory (Petrenko et al., 2020) framework, which reduces the wall time of a run to approximately two hours (see Appendix E.1). All experiments are conducted on a dedicated compute node with a 72-core 3.2 GHz AMD EPYC 7F72 CPU and a single NVIDIA A100 GPU. We adopt Sample-Factory's default settings for our network configuration, PPO setup, and training processes. For a more detailed experimental setup and exact hyperparameters please refer to Appendix B.

### 5.1 BASELINE PERFORMANCE

Table 3 shows that **PPO** maximizes rewards irrespective of costs, setting upper bounds on reward and cost in all environments. **PPOCost** finds a reward-cost trade-off with no guarantee that costs stay below the threshold (note that we plot the original reward before cost deduction). We analyze scaling the cost factor in Appendix C.2. **PPOLag** closely adheres to safety thresholds, yet often yields the lowest rewards. The continuous adjustment of the Lagrangian multiplier causes fluctuations that prevent consistent constraint satisfaction. **PPOSauté** and **P3O** exceed the cost threshold in several environments, although P3O achieves noticeably higher rewards. Conversely, **PPOPID** consistently meets the constraints, most often outperforming other baselines, making it arguably the most effective method on HASARD. The training curves are presented in Appendix H.

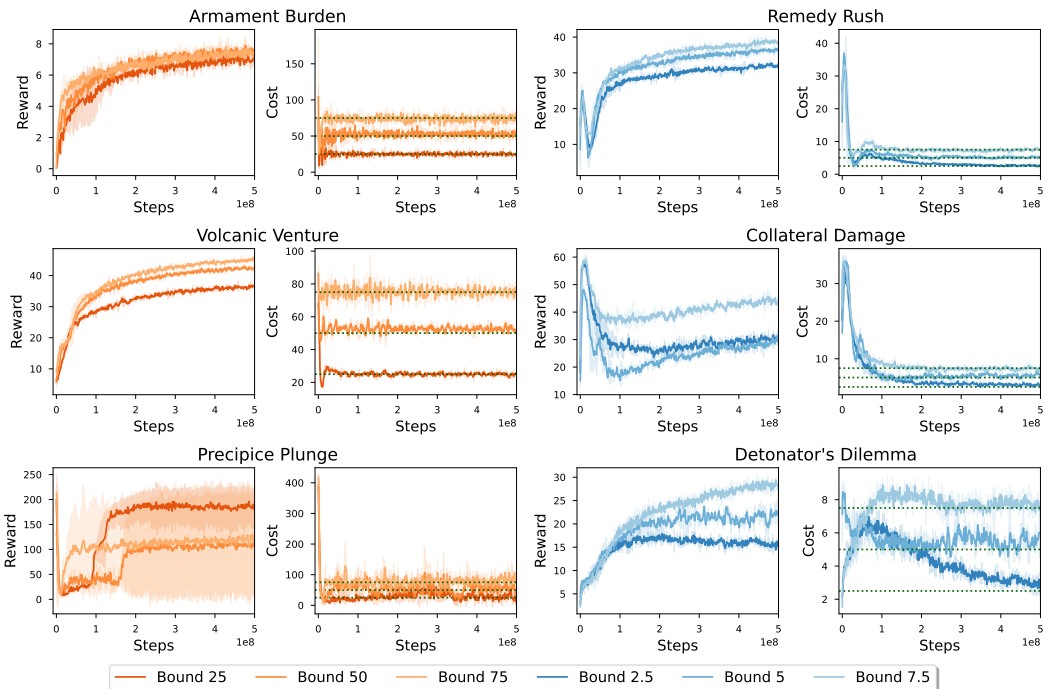

Figure 4: PPOLag's performance under varying safety budgets on Level 1. PPOLag consistently adheres to the set safety thresholds, with tighter cost limits yielding lower rewards.

## 5.2 VARYING SAFETY THRESHOLDS

The default safety bounds of HASARD environments offer a complex challenge with some margin for error. As safety requirements vary in real-world applications, these bounds are adjustable. To explore the effects of this, we run PPOLag on Level 1 with a higher and lower cost budget. Figure 4 shows that stricter cost limits result in lower rewards, although in some tasks the reward changes are disproportionate to the adjustments in the safety threshold.

## 5.3 AN IMPLICIT LEARNING CURRICULUM

Higher levels in HASARD not only feature parameter adjustments to increase difficulty but also introduce new mechanics. When trained on the hardest level directly, the agent often struggles to learn a good policy. Training RL agents on progressively more complex conditions to enhance learning efficiency has been widely explored (Florensa et al., 2017; Fang et al., 2019). To investigate whether starting with easier tasks can lead to better policies in HASARD, we evenly distribute the training budget across difficulty levels. We train PPOPID

Table 4: Difficulty levels of some HASARD environments provide a successful curriculum to learn a safe policy more efficiently.

| Training Regime | Remedy Rush | | Collateral Damage | |
|---|---|---|---|---|
| | R ↑ | C ↓ | R ↑ | C ↓ |
| Regular | 6.65 | 3.89 | 11.46 | 5.29 |
| Curriculum | 18.82 | 4.90 | 15.32 | 5.22 |

sequentially for 100M timesteps per level and compare the performance with an agent directly trained on Level 3 for 300M timesteps. We select the `Remedy Rush` and `Collateral Damage` environments because PPOPID exhibits significant performance gaps between levels, indicating the great potential to benefit from the competencies developed in easier tasks. Table 4 shows a nearly threefold performance increase in `Remedy Rush` and a 33% improvement in `Collateral Damage` over an equal number of timesteps. We find that leveraging the implicit curriculum from difficulty levels eases exploration challenges, prevents overly cautious behavior, and enables the agent to acquire skills that direct training fails to develop. Research on integrating curriculum learning with safe RL remains sparse. Previous studies (Eysenbach et al., 2017; Turchetta et al., 2020) have explored

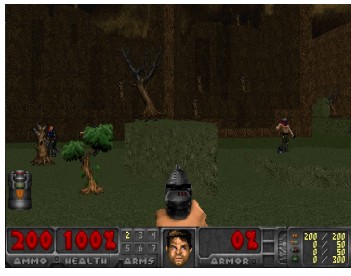 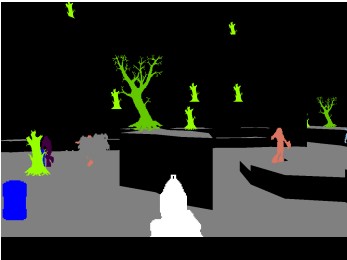 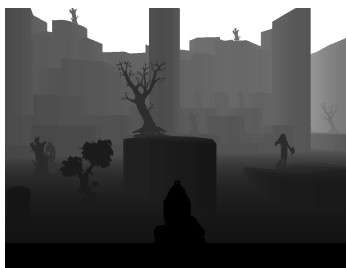

(a) Accurately assessing spatial relationships between objects and entities can be difficult to infer from **raw observations**.

(b) The **segmented observation** assigns each pixel a predefined color based on its label: item, unit, obstacle, wall, or surface.

(c) The **depth buffer** assigns each pixel a value from 0 (closest) to 255 (farthest). Intermediate values correspond to relative distances.

Figure 5: To analyze the visual complexity of HASARD, we create simplified representations through two strategies: (1) segmenting the observation and (2) including depth information.

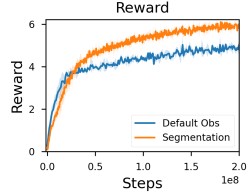 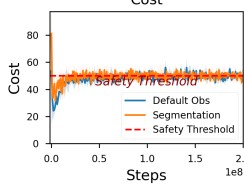 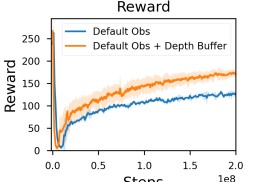 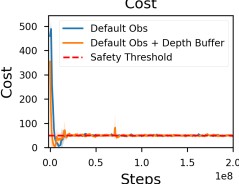

(a) **Segmentation Boost**: Training PPOPID on segmented observations in `Armament Burden` Level 2 yields a 21% reward increase with similar costs, highlighting the benefit of reduced visual complexity.

(b) **Depth Integration**: Augmenting the RGB input with depth information in `Precipice Plunge` Level 2 results in a 34% reward increase, underscoring the importance of accurate depth perception.

Figure 6: Impact of visual augmentations on learning.

this integration using specialized resetting agents to manage task difficulty and safety. HASARD's inherent curriculum across difficulty levels opens new avenues for exploration in this field.

## 5.4 VISUAL COMPLEXITY

Noisy visuals make it more difficult for RL agents to focus on the important aspects of the observations and effectively learn the task Hansen et al. (2021). While prior methods attempt to mask away the noise Bertoin et al. (2022); Grooten et al. (2023), we simply create simplified visual representations of the input observation by augmenting it. We investigate whether the agent has an easier time learning the task from the augmented representations and determine whether the visuals of HASARD environments are visually complex and observations noisy. Figure 5 depicts the visual augmentations.

**Segmentation** ViZDoom provides a labels buffer with ground-truth labels for each pixel. We use these labels to map each unique object to a fixed color to semantically segment the scene while ensuring consistent observations across frames. We hypothesize that this could help the agent perceive spatial relationships and better detect walls, surfaces, objects, and entities. We evaluate PPOPID on segmented observations in Armament Burden Level 2, chosen for its varied textures and numerous objects (7 weapon types and 4 obstacle types, plus walls, surfaces, and the agent, totaling 14 unique labels). Figure 6a shows a 21% reward increase with similar costs, indicating that segmentation simplifies learning.

**Depth Buffer** ViZDoom provides a depth buffer that assigns each pixel a value from 0 to 255, where 0 (black) indicates the closest objects and 255 (white) the farthest. Concatenating this additional channel to the RGB observation allows the agent to perceive spatial proximity directly. We evaluate PPO on `Precipice Plunge` Level 2, where precise depth estimation plays a key role in safely descending the cave. As shown in Figure 6b, this modification resulted in a 34% reward increase, underscoring the challenges of learning accurate depth perception from default observations alone.

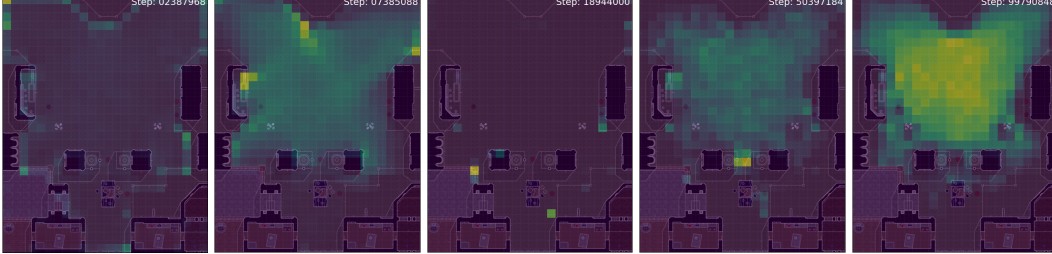

(a) In the early stages of training the agent is randomly exploring. This leads to frequent collisions with surrounding walls.

(b) The agent learns to navigate the map with a risky strategy, leading to high rewards at the expense of elevated costs.

(c) To reduce costs beneath the safety threshold, the agent adopts a conservative strategy by limiting its movements.

(d) Refinement of the safe strategy continues as the agent optimizes for higher rewards while maintaining low costs.

(e) The agent converges on a policy that consistently achieves high rewards without exceeding the cost budget.

Figure 7: Heatmaps of visited location frequency superimposed on the 2D map of `Remedy Rush`, illustrating PPOPID's policy evolution at key moments over 100M timesteps training on Level 1.

## 5.5 AGENT NAVIGATION

HASARD enables tracking the agent's visited locations during training. We aggregate the data from the 1000 most recent episodes and overlay it as a heatmap on the 2D environment map. This visual representation of the agent's movement patterns and exploration strategies provides insight into how the agent learns to solve a task. Figure 7 depicts the progressive refinement of the PPOPID policy on `Remedy Rush`. Early stages show random exploration with frequent wall collisions, followed by greedy reward maximization, then a shift toward an overly cautious strategy, and finally a convergence on a refined safe strategy that achieves high rewards while keeping costs within limits. This analysis provides a qualitative basis to compare how different methods attempt to learn the task. We present further examples in Appendix C.5.

## 6 CONCLUSION

Training competent agents able to navigate complex 3D tasks in safety-constrained settings remains a challenge in RL. HASARD stands as a useful and cost-effective tool for embodied vision-based learning, extending beyond mere navigation tasks. Our experiments reveal that current state-of-the-art safe RL methods struggle to learn complex behaviors under safety constraints. We anticipate HASARD to facilitate the development of future methods, offering valuable insights and novel problems to solve. Its adjustable difficulty system allows it to remain a valuable asset as safe RL methods evolve, contributing to safer autonomous systems that interact with the physical world.

## 7 LIMITATIONS AND FUTURE WORK

The environments, though complex and visually diverse, are based on the ViZDoom game engine. Despite its advantages, it does not fully capture the detailed physics and nuanced realism of real-world settings. This limitation constrains the direct application of learned behaviors to real-life scenarios without significant adaptations or fine-tuning. Most actions for interaction in Doom are discrete, inherently simplifying challenges of accurate control. Only the full action space can effectively be used for multi-task learning across environments to develop a general agent capable of mastering all scenarios. Our environments only have a single objective and safety constraint. Future work could include multiple safety constraints, multi-objective tasks, collaborative multi-agent scenarios, and continual or transfer learning scenarios. We discuss these directions more in-depth in Appendix G.

### ACKNOWLEDGMENTS

This work was conducted with the assistance of the Dutch national e-infrastructure, generously supported by the SURF Cooperative under grant EINF-8880. Additionally, this research was partially supported by the KOIOS project under grant agreement 101103770.

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

## A  ENVIRONMENT SPECIFICATIONS

In this section, we provide further details of HASARD's environments.

### A.1  DESCRIPTIONS AND MECHANICS

**Detonator's Dilemma**    The agent begins with 100 health points and is equipped with a pistol and unlimited ammunition to detonate the explosive barrels in the environment. Detonating a barrel grants a +1 reward. A single bullet is sufficient to detonate a barrel, but missed shots may hit nearby neutral units. Given the barrels' fragility, chain reactions are easy to occur. An example is depicted in Figure 8. Barrel explosions close to units will harm them. When a unit's HP reaches 0, it is eliminated, and the agent receives a penalty of -1. The agent is also penalized for harming itself, meaning that it should keep its distance when detonating the barrels. On elimination, a unit respawns at a random location, so the number of entities remains constant. Some creatures carry weapons that can be collected upon their elimination and can be used to detonate barrels more efficiently compared to the pistol. This can be employed as a sacrificial strategy to destroy more barrels. Two neutral units of each type are spawned at the beginning of an episode at random locations throughout the map. The units vary in resilience: `LostSoul` (10 HP), `ZombieMan` (25 HP), `ShotgunGuy` (40 HP), `ChaingunGuy` (55 HP), `DoomImp` (70 HP), `Demon` (85 HP), and `Revenant` (100 HP). Level 1 only includes `ShotgunGuy`, `DoomImp`, and `Revenant`. Level 2 adds `LostSoul` and `ChaingunGuy`, while Level 3 incorporates all the listed creatures. The environment features seven designated patrol points, to which after every five seconds, each unit is randomly assigned one to navigate towards. The patrol points are depicted in Figure 9.

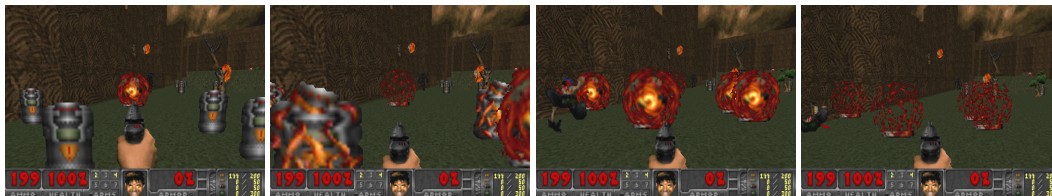

Figure 8: Detonating a barrel may cause a chain explosion of adjacent barrels. The impact thrusts the agent backward and causes severe neutral casualties.

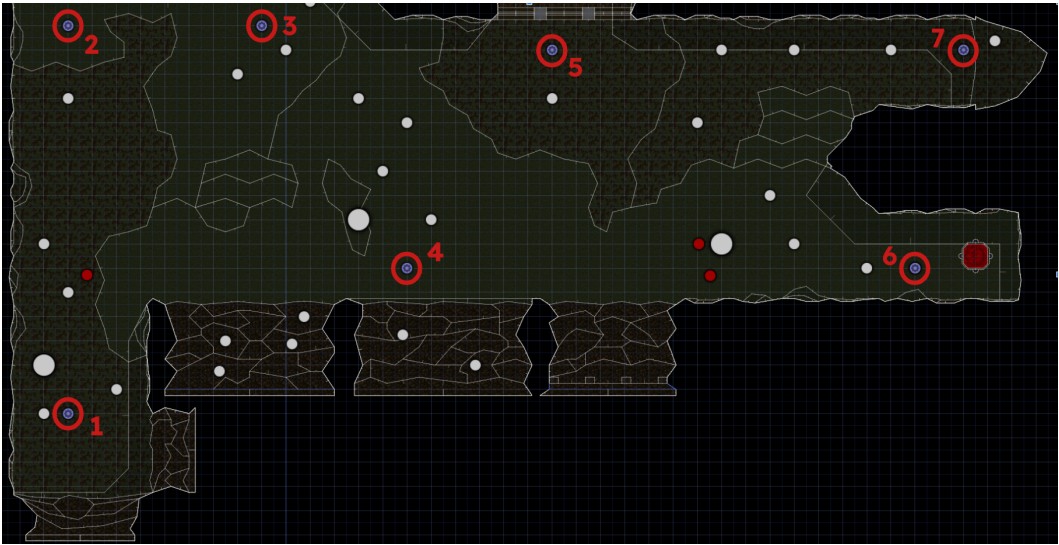

Figure 9: Patrol points of `Detonator's Dilemma`. After every 5 seconds, each neutral unit is randomly assigned one of the seven depicted locations to navigate toward. This strategy ensures that units are almost constantly in movement, and makes their movement patterns more predictable.

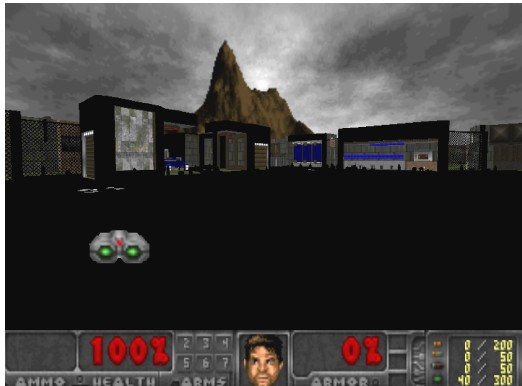 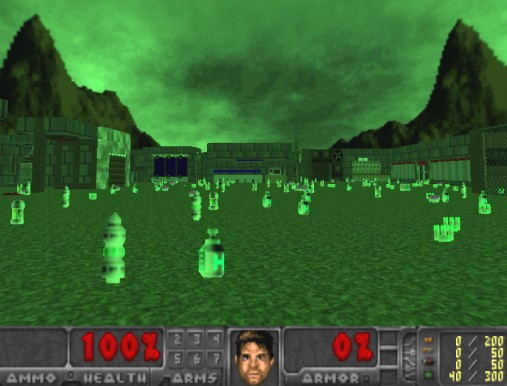

(a) During the darkness interval, both health and penalty items become imperceptible. The agent has a few strategies to consider: it could choose to remain stationary, avoiding the risk of collecting harmful items, or it could memorize the positions of desirable items and navigate by memory. Alternatively, the agent can seek out night vision goggles, which remain visible on the ground even in complete darkness, to maintain the ability to discern the items.

(b) Once the agent acquires the night vision goggles, it gains permanent visibility, unaffected by the darkness. However, the goggles render all items and surroundings less distinguishable by casting a pervasive green hue. While this change in color may seem minor to human players, it significantly affects image-based learning agents. Alterations in color, saturation, or hue necessitate learning to act in diverse conditions. This introduces an interesting strategic trade-off.

Figure 10: Levels 2 and 3 of `Remedy Rush` impose an additional navigation challenge. The light in the main sector is periodically switched off.

**Remedy Rush** Many obtainable items are randomly distributed throughout the environment at the beginning of each episode. The agent's objective is to collect health granting items $D^+ = \{\text{HealthBonus}, \text{Stimpack}, \text{Medikit}\}$ and avoid penalty items $D^- = \{\text{ArmorBonus}, \text{RocketAmmo}, \text{Shell}, \text{Cell}\}$. Additional items are spawned at random locations after every 120 in-game ticks ($\sim 3.5s$): two HealthBonus and one of each type from $D^-$. A new `Stimpack`, `Medikit`, and `Infrared` are spawned when picked up. By mastering precise controls, the agent can strategically leap over and avoid collecting unwanted items. In levels 2 and 3, the lighting of the environment alternates periodically between full brightness and complete darkness, adding a layer of difficulty as items become temporarily invisible. This effect is illustrated in Figure 10a. The agent can find night vision goggles randomly spawned within the level, which allow for uninterrupted visibility despite the fluctuating lights, though they cast a strong green hue on the surroundings (Figure 10b). Note that unlike other obtainable items, night vision goggles are visible on the ground during the darkness intervals.

**Armament Burden** At the start of an episode, 10 random weapons are spawned at random locations. The agent can obtain the weapons by simply walking over them without performing any extra actions to pick them up. The agent's movement speed $v$ is dynamically adjusted based on the total weight $w$ of the weapons carried. If $w$ exceeds the carrying capacity $c$, the agent's speed is modified according to the formula:

$$v = \max\left(0.1 \cdot v_0, v_0 - \frac{w - c}{c} \cdot v_0\right)$$

where $v_0$ is the agent's initial speed. This mechanism ensures that the speed reduction is proportional to the excess weight but does not drop below 25% of $v_0$, thereby preventing total immobility. When the agent reaches the delivery zone, it regains its original movement speed $v_0$. Simultaneously, the agent receives a reward based on the types and quantities of weapons delivered. Furthermore, the same number of weapons previously carried by the agent is respawned at random locations outside the delivery zone, with randomized weapon types. The agent can discard all carried weapons by utilizing the `USE` action. This can be strategically used to 1) lighten the load and restore the original movement, 2) avoid further penalties if the carrying load has been breached, 3) preemptively remove decoy items at key locations where they substantially inhibit the agent's movement, and 4) eliminate the weapons at the back of the area far from the delivery zone, hoping they would respawn closer. The latter two strategies impose a potential opportunity cost, as the time spent on discarding could be

Table 5: Obtainable items found in the `Armament Burden` scenario with their associated delivery rewards and weights.

| Item | Pistol | Shot-gun | Super Shot-gun | Chain-gun | Rocket Launcher | Plasma Rifle | BFG 9000 | Blur Sphere | All map | Back pack | Rad Suit |
|---|---|---|---|---|---|---|---|---|---|---|---|
| **Weight** | 0.1 | 0.25 | 0.4 | 0.55 | 0.7 | 0.85 | 1.0 | 0.25 | 0.5 | 0.75 | 1.0 |
| **Reward** | 0.1 | 0.25 | 0.4 | 0.55 | 0.7 | 0.85 | 1.0 | 0 | 0 | 0 | 0 |

used to maximize rewards by delivering weapons. However, sacrificing this time could yield higher returns later in the episode. Higher difficulty levels introduce new features. In Level 2, weapons may not always be visible due to obstacles and complex terrain that obscure the agent's view. The agent can navigate more effectively with the use of the `JUMP` action. Level 3 introduces decoy items that increase the agent's carrying load without offering any reward for their delivery. This creates a challenge in credit assignment, as the agent cannot implicitly discern which items picked up contributed to the delivery reward. Table 5 displays the rewards and weights of all weapons and decoys. Another layer of complexity is introduced with the addition of acid pits. Each episode features 20 acid pits of a fixed size, randomly placed throughout the level. Falling into one of these pits results in the agent losing all its health, accompanied by a significant cost penalty.

**Volcanic Venture**   The agent has to collect items in an environment, where a percentage of random floor tiles are covered with lava. `ArmorBonus` collectible items are randomly distributed across the map, which may also appear on the lava tiles. An additional item is spawned after every 60 in-game ticks ($\sim 2s$). The agent starts with $H_0 = 1000$ health points. Stepping on lava results in a loss of 1 health point per timestep. Higher difficulty levels have less of the surface area covered by platforms. Platform heights vary in Levels 2 and 3, and their locations change after a fixed time interval. When this occurs, the agent is granted a short period of invulnerability to ensure that health loss and cost can entirely be avoided. Figure 11 further describes the complexity introduced by terrain variations across levels. With hard constraints, the agent suffers a severe penalty by losing all remaining health upon contact with lava, instantly ending the episode.

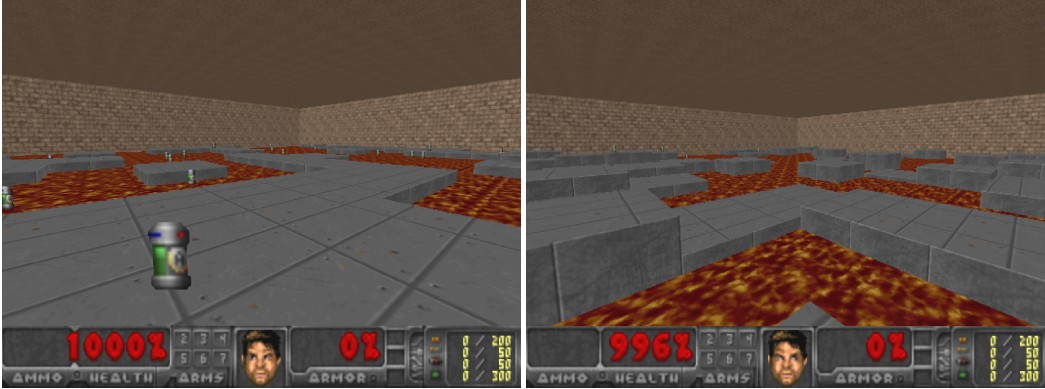

(a) Level 1 features a layout where the environment lava surface is partially covered with platforms, which remain at fixed positions throughout an episode. The agent's objective is to collect items that spawn continuously while avoiding contact with the lava.

(b) In higher levels, platform heights vary, creating a more challenging terrain to navigate. Their locations also change at regular intervals. Level 3 introduces further complexity by sporadically waggling the platforms up and down.

Figure 11: Difficulty Levels in `Volcanic Venture` progressively increase complexity with dynamic and non-stationary terrain features.

**Collateral Damage**   The agent is armed with a `RocketLauncher` capable of firing a rocket every eight frames. The task is to eliminate hostile units (`Cacodemon`) without harming neutral units (`Zombieman`). The units spawn at random locations on the other side of the environment away from the agent. A unit immediately respawns after being eliminated. The rocket's high area of effect (AoE) explosion presents an extra safety challenge, as illustrated in Figure 12. Due to the projectile's travel

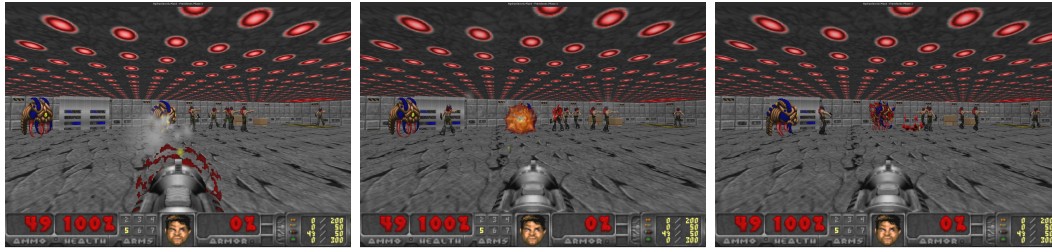

(a) The agent launches a rocket directly towards an enemy unit. (b) The target unit takes a direct hit from the rocket. (c) An adjacent neutral unit is harmed by the explosion.

Figure 12: The explosion from the rocket has a high area of effect (AoE), potentially causing casualties even if the enemy unit has taken a direct hit.

time across the environment, there is a delay before impact, during which the entities' positions may change. The agent's distance from the units increases across difficulty levels. Higher levels further increase the challenge by featuring more neutral and less hostile units. Neutral units exhibit slow, random movements, adding an element of unpredictability. In contrast, enemy units move more swiftly between designated points on the left and right sides of the map from the agent's perspective. Hostile unit speed increases with the difficulty level. In this scenario, the agent cannot move or harm itself.

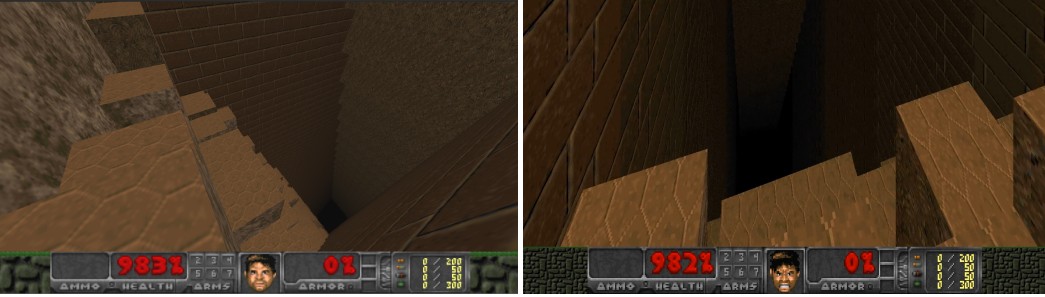

Figure 13: The ViZDoom game engine employs a rendering technique called "non-perspective correct texture mapping" for vertical adjustments, leading to noticeable distortions when the player looks up or down. This causes the textures and environment to stretch or squash, resulting in a fisheye effect (**right**). In contrast, the more advanced GZDoom engine accurately renders views for vertical perception, avoiding such distortions (**left**).

**Precipice Plunge** In Level 1 of the task, the agent must descend the cave by a winding staircase formed by adjacent pillars, with each step having 24 vertical units lower than the previous. In Levels 2 and 3, the design changes: each row of pillars is positioned 192 units lower than the one above, and the heights of individual pillars are randomized by $\pm 50$ units, so the path down no longer resembles a staircase. In Level 3, the pillars further oscillate vertically—moving up and down by a random amount between 128 and 512 in-game units at a random *waggle frequency* between 12 and 24, where higher values produce faster oscillations. As the agent descends deeper into the cave, the environment becomes progressively darker. The starting vertical height for the agent is denoted as $h_0 = 0$. For each subsequent row of platforms $k$, the vertical height decreases by a fixed amount $\Delta$, with the actual platform height at row $k$ being $h_k = -k \cdot \Delta + rand\left[-\frac{\Delta}{2}, \frac{\Delta}{2}\right]$, where $rand[\cdot, \cdot]$ generates a random variation within this range. The agent must avoid fall damage $D$, which is calculated by

$$D = \begin{cases} (d - \theta) \cdot \alpha, & \text{if } d > \theta \\ 0, & \text{if } d \leq \theta \end{cases}$$

where $d$ is the vertical distance fallen by the agent, $\theta = 96$ is the threshold for fall damage (below which no damage occurs), and $\alpha = 0.1$ is the damage multiplier. It's important to note that the ViZDoom engine does not effectively support vertical aiming (looking upwards or downwards), resulting in heavy distortions, as illustrated in Figure 13. Although the ability to look up and down is essential for effectively completing the task, it introduces visual compromises due to the limitations of the engine.

## A.2 REWARD FUNCTIONS

In this section, we define how rewards $R(t)$ are incurred at any given time step $t$ in each environment.

**Armament Burden**    $R(t) = \sum_{i \in W_t} r_i$, where $W_t$ represents the set of weapons picked up at time $t$, and $r_i$ is the reward associated with each weapon $i$ picked up. The rewards for weapons are $R_w = \{0.1, 0.25, 0.4, 0.55, 0.7, 0.85, 1.0\}$, corresponding to the collection {Pistol, Shotgun, SuperShotgun, Chaingun, RocketLauncher, PlasmaRifle, BFG9000}.

**Remedy Rush**    $R(t) = r_1 \cdot v_t + r_2 \cdot s_t + r_3 \cdot m_t$, where $r_1 = 1.0, r_2 = 3.0, r_3 = 6.0$ are the reward scalars for vials $n_t$, stimpacks $s_t$ and medikits $m_t$ collected at time $t$.

**Collateral Damage**    $R(t) = r \cdot n_t$, where $r = 1.0$ is the reward for eliminating an enemy and $n_t$ is the number enemies eliminated at time $t$.

**Volcanic Venture**    $R(t) = r \cdot n_t$, where $r = 1.0$ is the reward for collecting an armor bonus and $n_t$ is the number of resources collected at time $t$.

**Precipice Plunge**    $R(t) = \alpha \cdot \max(0, z_{t-1} - z_t)$, where $\alpha = 0.05$ is a positive constant scaling the reward for each unit of depth reached, $z_t$ is the current $z$-coordinate of the agent, and $z_{t-1}$ is the previous one.

**Detonator's Dilemma**    $R(t) = r \cdot n_t$, where $r = 1.0$ is the reward for detonating a barrel and $n_t$ is the number of barrels exploded at time $t$.

## A.3 COST FUNCTIONS

In this section, we define how cost $C(t)$ is incurred at any given time step $t$ in each environment.

**Armament Burden**    $C_{soft}(t) = (1 + (\rho - 1) \cdot \mathbb{1}(\omega_t = 0)) \cdot \frac{\max(0, \sum_{i \in W_t} w_i - c)}{c}$, where $W_t$ is the collection of weapons carried at time step $t$, $w_i$ denotes the weight of weapon $i$, $\mathbb{1}(\cdot)$ is an indicator function, $\omega_t$ is a binary variable indicating whether a weapon is obtained at time $t$, $\rho = 0.1$ is the penalty coefficient for carrying excess weight, and $c$ is the carrying capacity. The weights of weapons are $\mathcal{W} = \{0.05, 0.15, 0.3, 0.6, 1.0, 3.0, 6.0\}$, corresponding to the collection {Pistol, Shotgun, SuperShotgun, Chaingun, RocketLauncher, PlasmaRifle, BFG9000}. In the hard constraint scenario, we employ $C_{hard}(t) = H \cdot \mathbb{1}(\sum_{i \in W_t} w_i > c)$, where $H = 10$ represents the hard constraint penalty applied instantaneously when the weight of the weapons exceeds the carrying capacity.

**Remedy Rush**    $C(t) = \sum_{i \in I_t} \mathbb{1}(i_t \in D^-)$, where $I_t$ is the set of items obtained at timestep $t$, $D^-$ is the set of incorrect items, and $\mathbb{1}(\cdot)$ is an indicator function.

**Collateral Damage**    $C(t) = n_t$, where $n_t$ is the number neutral entities eliminated at time $t$.

**Volcanic Venture**    $C(t) = H_{t-1} - H_t$, where $H_t$ is the agent's health at time step $t$.

**Precipice Plunge**    $C(t) = H_{t-1} - H_t$, where $H_t$ is the agent's health at time step $t$.

**Detonator's Dilemma**    $C(t) = n_t + \alpha \cdot (H_{t-1} - H_t)$, where $n_t$ denotes the number of neutral entities eliminated at time $t$, $H_{t-1}$ and $H_t$ are the agent's health at the previous and current timesteps, respectively, and $\alpha = 0.04$ is the health penalty scaling factor.

## A.4 DIFFICULTY LEVEL ATTRIBUTES

Table 6 displays the difficulty attributes of each level.

Table 6: Level Difficulty Attributes of Environments. The attributes are explained in Appendix A.1

| Environment | Attribute | Level 1 | Level 2 | Level 3 |
|---|---|---|---|---|
| Armament Burden | Complex Terrain | ✗ | ✓ | ✓ |
| | Obstacles | ✗ | ✓ | ✓ |
| | Pitfalls | ✗ | ✗ | ✓ |
| | Decoy Items | ✗ | ✗ | ✓ |
| Remedy Rush | Health Vials | 30 | 20 | 10 |
| | Hazardous Items | 40 | 60 | 80 |
| | Darkness Duration | N/A | 20 | 40 |
| | Night Vision Goggles | N/A | 2 | 1 |
| Collateral Damage | Hostile Targets | 4 | 3 | 2 |
| | Target Speed | 10 | 15 | 20 |
| | Neutral Units | 4 | 5 | 6 |
| | Neutral Health | 60 | 40 | 20 |
| | Distance From Units | 256-456 | 400-600 | 544-744 |
| Volcanic Venture | Lava Coverage | 60% | 70% | 80% |
| | Changing Platforms | ✗ | ✓ | ✓ |
| | Random Platform Height | ✗ | ✓ | ✓ |
| | Platforms Waggle | ✗ | ✗ | ✓ |
| Precipice Plunge | Step Decrement | 24 | 128 | 192 |
| | Darkness Fluctuation | 30 | 30 | 50 |
| | Randomized Terrain | ✗ | ✓ | ✓ |
| | Moving Pillars | ✗ | ✗ | ✓ |
| Detonator's Dilemma | Creature Types | 3 | 5 | 7 |
| | Creature Speed | 8 | 12 | 16 |
| | Explosive Barrels | 10 | 15 | 20 |

## B EXPERIMENTAL SETUP

We adopt most of our experimental setup from the PPO implementation for ViZDoom environments in Sample-Factory (Petrenko et al., 2020).

### B.1 NETWORK ARCHITECTURE

The pixel observations from the environment are first processed by a CNN encoder from (Mnih et al., 2015), incorporating ELU for nonlinearity. The convolutions are passed through two dense layers, each with 512 neurons. A GRU with 512 hidden units processes the sequential and temporal information from the environment. Afterward, the architecture splits into an actor and critic network, sharing the same backbone. The actor produces a categorical distribution of action probabilities for each single action set, while the critic outputs a scalar value estimate for state-action pairs to guide policy improvements.

### B.2 HYPERPARAMETERS

We present the extensive list of hyperparameters used in our experiments in Table 7. The hyperparameters are shared between our baselines.

### B.3 IMPLEMENTATION DETAILS

Each action selected by the agent's policy is repeated for four frames, reducing the need for frequent policy decisions and conserving computational resources. During preprocessing, the RGB obser-

| Parameter | Value | Description |
|---|---|---|
| Batch Size ($B$) | 1024 | Minibatch size for SGD |
| Gamma ($\gamma$) | 0.99 | Discount factor |
| Learning Rate ($\alpha$) | $1 \times 10^{-4}$ | Learning rate |
| Hidden Layer Sizes | $512, 512$ | Number of neurons in the dense layers after the convolutional encoder |
| RNN | GRU | Type of the RNN |
| RNN Size ($h$) | 512 | Size of the RNN hidden state |
| Policy Init Gain ($g$) | 1.0 | Gain parameter of neural network initialization schemas |
| Exploration Loss Coeff. ($C_{expl}$) | 0.001 | Coefficient for the exploration component of the loss function |
| Value Loss Coeff. ($C_{val}$) | 0.5 | Coefficient for the critic loss |
| Lambda Lagrange ($\lambda_{lagr}$) | 0.0 | Lambda coefficient for the Lagrange multiplier |
| Lagrangian Coeff. Rate ($r_{lagr}$) | $1 \times 10^{-2}$ | Change rate of the Lagrangian coefficient |
| KL Threshold ($\theta_{KL}$) | 0.01 | Threshold for the KL divergence between the old and new policy |
| GAE Lambda ($\lambda_{GAE}$) | 0.95 | Generalized Advantage Estimation discounting |
| PPO Clip Ratio ($\epsilon_{clip}$) | 0.1 | PPO clipping ratio, unbiased clip version |
| PPO Clip Value ($\Delta_{clip}$) | 1.0 | Maximum absolute change in value estimate until it is clipped |
| Nonlinearity ($\phi$) | ELU | Type of nonlinear activation function used in the network |
| Optimizer | Adam | Type of the optimizer |
| Adam Epsilon ($\epsilon_{Adam}$) | $1 \times 10^{-6}$ | Adam epsilon parameter |
| Adam Beta ($\beta_1, \beta_2$) | 0.9, 0.999 | Adam first and second momentum decay coefficient |
| Max Grad Norm | 4.0 | Max L2 norm of the gradient vector |
| Policy Initialization | orthogonal | Neural network layer weight initialization method |
| Frame Skip | 4 | Number of times to repeat a selected action in the environment |
| Frame Stack | 1 | Number of consecutive environment pixel-observation to stack |
| Env Workers | 32 | Number of parallel environment CPU workers |
| Num Envs per Worker | 10 | Number of environments managed by a single CPU actor |
| Accumulate Batches | 2 | Max number of training batches the learner accumulates before stopping |
| Worker Num Splits | 2 | Enable double buffered experience collection, vector environment splits |
| Policy Workers per Policy | 1 | Number of workers that compute the forward pass for each policy |
| Max Policy Lag | 1000 | Beyond how many steps to discard older experiences |

Table 7: Hyperparameters

vations are downscaled from $160 \times 120$ pixels to $128 \times 72$ pixels. We accumulate a maximum of two training batches at a time, preventing data collection from outpacing training and maintaining a balance between throughput and policy lag. We split an environment into two for a double-buffered experience collection. Each policy has a designated worker responsible for the forward pass, and we maintain a policy lag limit of 1000 steps to ensure data relevance. Parallel environment workers are dynamically set based on CPU availability. The training cycle involves collecting batches to form a dataset size defined by the product of batch size and the number of batches per epoch, with a standard batch size set at 1024. Rollouts are conducted over 32 timesteps, aligned with the recurrence interval necessary for RNN policies, facilitating efficient data processing and learning accuracy.

## C  EXTENDED BENCHMARK ANALYSIS

This section offers a more comprehensive examination of our baseline methods and the insights enabled by HASARD. We investigate how the full action space affects learning, assess the impact of hard safety constraints, explore PPOCost's sensitivity to cost scaling, compare baseline performances using different RL algorithms, and analyze how agents' navigation strategies evolve across different tasks.

### C.1  RL BASELINE COMPARISON

All of our selected baselines in the main paper are extensions of PPO for safety. It is important to note that the selected Safe RL methods are inherently independent of any particular base RL algorithm. However, in the original papers, the authors frequently utilized PPO as a base for their model-free methods. We therefore only included the PPO-based version not only because those have achieved higher results on prior benchmarks, but also not to compare apples with oranges, i.e., not to conflate differences in performance due to the choice of the base RL algorithm.

To demonstrate that the benchmark is not tied to a particular type of base RL algorithm, we include TRPO-based Schulman et al. (2015) Safe RL method implementations which were included in the original papers. Table 8 presents the results of Level 1. Although TRPO-based methods meet the safety threshold, they consistently yield lower rewards—mirroring the performance gap observed between the base PPO and TRPO implementations.

Table 8: Comparison of safety algorithms implemented on **TRPO** and **PPO**. The green values indicate results satisfying the safety threshold, while **bold** values highlight the best reward achieved while staying within the cost bound. If no method meets the safety threshold, the result with the lowest cost is shown instead. Results on Level 1 show that the PPO-based implementations consistently outperform their TRPO-based counterparts.

| Method | Armament Burden | | Volcanic Venture | | Remedy Rush | | Collateral Damage | | Precipice Plunge | | Detonator's Dilemma | |
|---|---|---|---|---|---|---|---|---|---|---|---|---|
| | R↑ | C↓ | R↑ | C↓ | R↑ | C↓ | R↑ | C↓ | R↑ | C↓ | R↑ | C↓ |
| PPO | 9.68 | 109.30 | 51.64 | 172.93 | 50.78 | 52.44 | 78.61 | 41.06 | 243.42 | 475.62 | 29.67 | 14.28 |
| TRPO | 7.24 | 144.79 | 41.00 | 185.30 | 45.70 | 48.47 | 67.91 | 37.29 | 243.41 | 468.99 | 20.03 | 8.92 |
| PPOLag | 7.51 | 52.41 | 42.40 | 52.00 | 36.37 | 5.25 | 29.09 | 5.61 | 147.24 | 44.96 | 21.49 | 5.62 |
| TRPOLag | 4.06 | 48.61 | **30.48** | 49.91 | 21.69 | 4.98 | 31.63 | 7.28 | 177.93 | 309.16 | 16.84 | 6.85 |
| PPOPID | **8.99** | 49.79 | 45.23 | 50.53 | **38.19** | 4.90 | **43.27** | 5.03 | **231.53** | 43.91 | **26.51** | 5.25 |
| TRPOPID | 0.36 | 8.18 | 23.81 | 50.93 | 10.36 | 4.54 | 34.74 | 10.13 | 172.87 | 299.46 | 12.81 | 6.84 |

### C.2  COST FACTOR SENSITIVITY

The PPOCost baseline, while simple and straightforward in approach, has demonstrated its potential in our experiments. On Level 1 of `Precipice Plunge`, it achieved nearly zero cost while securing the highest reward among methods that adhere to the safety threshold. Similarly, in Level 2 of `Armament Burden`, PPOCost again obtained the highest reward among all methods compliant with the given budget. However, its effectiveness hinges on determining the appropriate cost scaling factor, which requires manual tuning through a time-consuming and costly trial-and-error process. Consequently, we explore how sensitive PPOCost is to variations in the penalty scaling factor. We evaluate the cost scaling values of [0.1, 0.5, 1.0, 2.0] on Level 1 of all environments. Note that in the main experiments, we arbitrarily chose a coefficient of 1.0. We present the training curves in Figure 14 and the evaluation results in Table 9. We can observe that the reward and cost are tightly bound: a reduction of cost necessitates a reward sacrifice. Note, that we introduced PPOCost as a proof of concept to demonstrate this direct trade-off in our environments along with the extent to which cost can function as a negative reward on the benchmark.

Table 9: Performance metrics of PPOCost across varying cost scales.

| Cost Scale | Armament Burden | | Volcanic Venture | | Remedy Rush | | Collateral Damage | | Precipice Plunge | | Detonator's Dilemma | |
|---|---|---|---|---|---|---|---|---|---|---|---|---|
| | $R\uparrow$ | $C\downarrow$ | $R\uparrow$ | $C\downarrow$ | $R\uparrow$ | $C\downarrow$ | $R\uparrow$ | $C\downarrow$ | $R\uparrow$ | $C\downarrow$ | $R\uparrow$ | $C\downarrow$ |
| 0.1 | 16.40 | 9.09 | 128.61 | 92.23 | 94.94 | 50.49 | 109.61 | 41.34 | 609.65 | 408.35 | 35.27 | 13.25 |
| 0.5 | 3.86 | 1.64 | 42.30 | 19.13 | 57.04 | 23.06 | 86.67 | 33.23 | 441.61 | 399.14 | 27.93 | 10.28 |
| 1.0 | 0.13 | 0.10 | 25.93 | 6.00 | 18.30 | 6.03 | 64.79 | 24.60 | 196.93 | 2.29 | 18.39 | 6.37 |
| 2.0 | 0.00 | 0.04 | 17.64 | 2.12 | 0.00 | 0.03 | 30.41 | 8.55 | 203.46 | 1.20 | 0.57 | 0.20 |

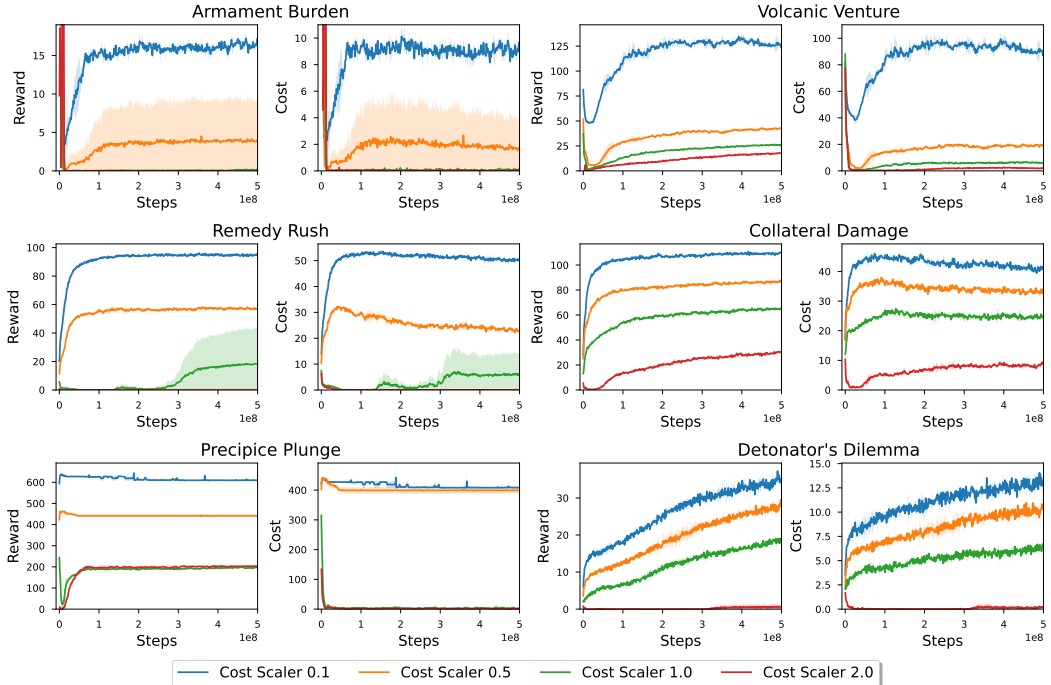

Figure 14: PPOCost treats costs as negative rewards without providing a general method for scaling these costs relative to rewards. Training with various cost scaling values indicates that PPOCost is highly sensitive to this parameter, resulting in a wide range of reward-cost trade-offs. For the main results of the paper, we adopted a scaling factor of 1.0, ensuring equal weight between rewards and negative costs. With extensive manual tuning, PPOCost can find diverse trade-offs, however, it cannot satisfy a given safety budget.

## C.3 HARD SAFETY CONSTRAINTS

Under hard constraints ($\xi = 0$), any action that violates safety leads to one of two outcomes: (1) termination of the episode, where the trajectory is deemed a failure, all rewards are withheld, and substantial costs are imposed. This approach redefines the episode's time horizon to $T = \min\{t \mid c_i(s_t, a_t) > 0\}$, effectively shortening the episode to the point of the first safety violation. Alternatively, (2) it results in severe in-game penalties that drastically reduce progress and potential outcomes. For instance, in `Armament Burden`, the agent loses all obtained weapons when exceeding the carrying capacity. Figure 15 illustrates the effects of soft and hard constraints in selected game scenarios, demonstrating the impact of each constraint type on gameplay dynamics and strategy. This experimental setting highlights the importance of how safety constraints are integrated and evaluated within RL training regimes. Whereas the default *soft* constraint setting assumes there is a cost budget in the bounds of which the agent can navigate, certain applications are more safety-critical. Dealing with the potential of crashing a car into a wall presents very different requirements. As there are several known formulations of safety in RL (Wachi et al., 2024b), it is important to acknowledge what type of safety problem an algorithm is designed to solve. With this

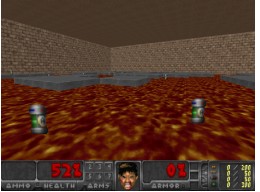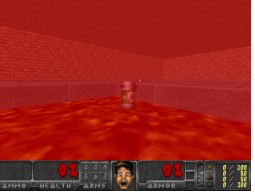 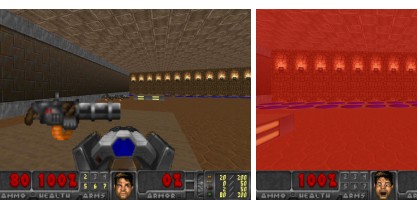

(a) `Volcanic Venture`: Under the **soft** constraint, the agent can step on the lava to collect items or reach other platforms, albeit at a health cost. Under the **hard** constraint, any contact with lava results in immediate termination of the episode due to total health loss.

(b) `Armament Burden`: Upon exceeding capacity the **soft** constraint slows the agent, allowing continued progress with reduced speed, whereas the **hard** constraint causes the agent to drop all weapons, forcing a restart of the collection process.

Figure 15: The hard constraint setting affects the agent's progression more severely.

setting, HASARD therefore aims to provide a versatile benchmark that facilitates evaluation across different safety formulations.

Table 10: Performance on Level 1 of the HASARD benchmark with hard constraints across five unique seeds. We show the average of the final ten data points. Maximum rewards and minimum costs are highlighted in **bold**.

| Method | Armament Burden | | Volcanic Venture | | Remedy Rush | | Collateral Damage | | Precipice Plunge | | Detonator's Dilemma | |
|---|---|---|---|---|---|---|---|---|---|---|---|---|
| | **R** ↑ | **C** ↓ | **R** ↑ | **C** ↓ | **R** ↑ | **C** ↓ | **R** ↑ | **C** ↓ | **R** ↑ | **C** ↓ | **R** ↑ | **C** ↓ |
| PPO | **13.97** | 0.70 | **15.47** | 1410 | **18.09** | 8.28 | **10.12** | 9.40 | 38.40 | 10.00 | **3.73** | 12.52 |
| PPOCost | 5.19 | 0.03 | 0.65 | 15.83 | 0.01 | 0.21 | 0.47 | 0.55 | 37.00 | 10.00 | 0.67 | 0.87 |
| PPOLag | 0.01 | **0.00** | 1.15 | **12.49** | 0.01 | **0.19** | 0.15 | **0.54** | **194.4** | **0.65** | 0.03 | **0.13** |

It has been shown that in environments with hard constraints, Safe RL agents suffer from the *safe exploration problem* (Garcia & Fernández, 2012; Pecka & Svoboda, 2014). We observe the same issue as indicated by the results in Table 10. The penalty for risky exploration is prohibitive, pushing PPOCost and PPOLag towards overly cautious behaviors, as evidenced by their near-zero rewards. `Precipice Plunge` is the only environment, where PPOLag manages to learn a safe policy, allowing it to navigate to the bottom of the cave without falling a single time. Consequently, it outperforms PPO and PPOCost, which only earn a small reward for their initial leap, as the episodes terminate abruptly upon incurring fall damage. Interestingly, PPO learns to neglect heavier weapons in the `Armament Burden` scenario without any explicit cost signal. This behavior appears to be a response to the lack of rewards associated with these weapons, as exceeding the carrying capacity results in the loss of all acquired weapons. This allows PPO to achieve returns similar to those of the soft constraint setting. A potential approach to developing a successful policy involves using a curriculum that progressively reduces the cost budget until it reaches zero. We leave this open for future research.

## C.4 FULL ACTION SPACE

The full action space incorporates the following array of actions: $D = \{$ MOVE_FORWARD, MOVE_BACKWARD, MOVE_RIGHT, MOVE_LEFT, SELECT_NEXT_WEAPON, SELECT_PREV_WEAPON, ATTACK, SPEED, JUMP, USE, CROUCH, TURN180, LOOK_UP_DOWN_DELTA, TURN_LEFT_RIGHT_DELTA $\}$. Some actions, such as USE, only have an effect in certain environments like `Armament Burden` and `Precipice Plunge`. In

Table 11: Action space comparison between simplified and full configurations.

| **Actions** | Discrete | Continuous |
|---|---|---|
| Simplified | 6–54 | 0 |
| Original | 864 | 2 |

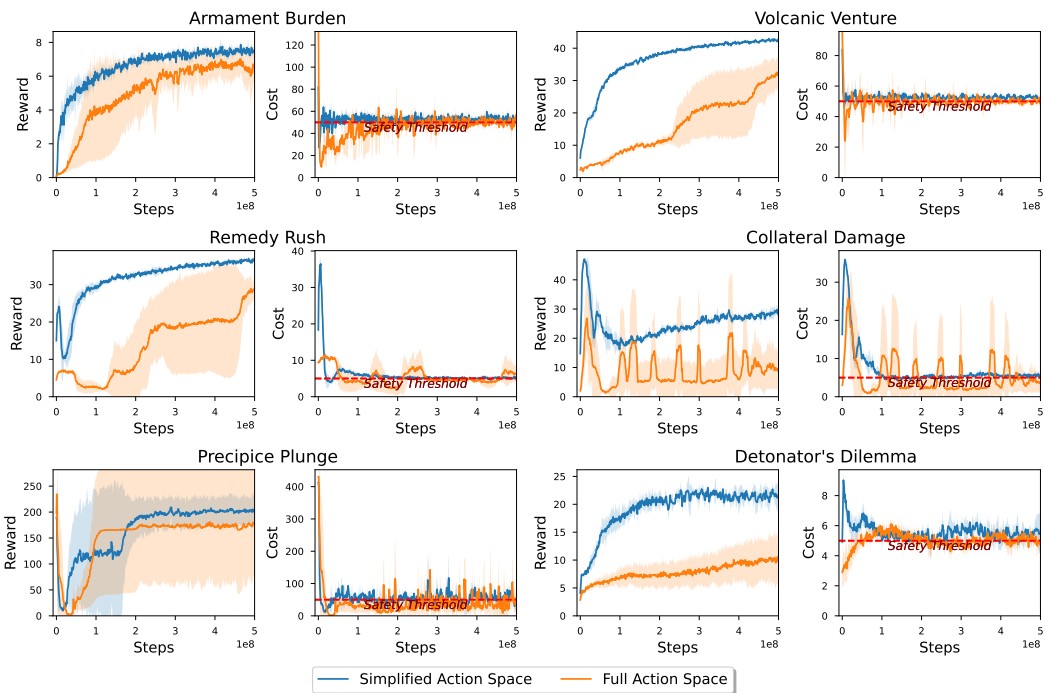

Figure 16: While the complete DOOM action space potentially facilitates more efficient task resolution with a wider array of actions at the disposal of the agent, PPOLag fails to leverage this advantage, even at Level 1 of HASARD. Although PPOLag consistently meets the default safety budgets across environments, it significantly lags behind in achieving rewards comparable to those obtained using a simplified action space. The full DOOM action space therefore presents a much more complicated Safe RL learning problem.

Table 12: The full action space setting presents a far greater challenge to PPOLag in Level 1.

| Action Space | Armament Burden | | Volcanic Venture | | Remedy Rush | | Collateral Damage | | Precipice Plunge | | Detonator's Dilemma | |
|---|---|---|---|---|---|---|---|---|---|---|---|---|
| | $R\uparrow$ | $C\downarrow$ | $R\uparrow$ | $C\downarrow$ | $R\uparrow$ | $C\downarrow$ | $R\uparrow$ | $C\downarrow$ | $R\uparrow$ | $C\downarrow$ | $R\uparrow$ | $C\downarrow$ |
| Simplified | 7.51 | 52.41 | 42.40 | 52.00 | 36.37 | 5.25 | 29.09 | 5.61 | 147.24 | 44.96 | 21.49 | 5.62 |
| Full | 4.97 | 52.74 | 26.43 | 49.58 | 24.96 | 9.26 | 14.03 | 4.85 | 40.84 | 76.41 | 4.98 | 4.42 |

other scenarios, such actions are redundant, adding an extra overhead for the agent to discern their irrelevance. Other actions could offer quicker alternatives for achieving certain objectives. For example, `MOVE_LEFT` directly allows sidestepping to the left, whereas a sequence of `TURN_LEFT` → `MOVE_FORWARD` → `TURN_RIGHT` accomplishes the same, however much slower. Therefore, in theory, each task could be solved more effectively with the full action space; however, its complexity presents a more challenging learning problem. To demonstrate this complexity, we run PPOLag on the easy level of all environments with the full action space. We present the evaluation results in Table 12 and the training curves in Figure 16. For a fair comparison with the original results, we restrict movement in `Collateral Damage` and acceleration in `Armament Burden`. PPOLag adheres to the default safety budget when using the full action space, but experiences a significant reduction in reward. We have made the use of the full action space a configurable option, allowing benchmark users to tailor it according to their needs.

## C.5 EXTENDED AGENT NAVIGATION

This section presents additional examples of insight into agent behavior during training via heatmap visualization of spatial visitations. There are numerous policies that, despite exhibiting distinct

behaviors, could result in the same reward and cost outcomes. This visualization offers a window into the qualitative evolution of agent behavior. We can observe how initially random exploration gradually gives way to more deliberate navigation patterns as the agent learns to balance reward-seeking with safety constraints. This not only validates performance metrics but helps in understanding how agents internalize the task objective and safety constraints. Such insights help diagnose potential inefficiencies or overly cautious behaviors in different algorithms and guide further refinement.

Figure 17) shows how PPO opts for a direct leap down the `Precipice Plunge` cave, while the PPOPID agent has learned to follow the winding path to mitigate fall damage. Figure 18 depicts how increasing difficulty levels inhibit the agent's movement from the delivery zone in `Armament Burden`. The heatmaps of `Detonator's Dilemma` in Figure 19 reveal that the agent prefers to cover different sections of the map during different stages of training as its policy evolves. Finally, Figure 20) compares the unsafe movement patterns of PPO in `Volcanic Venture` with the PPOPID agent optimized for safety.

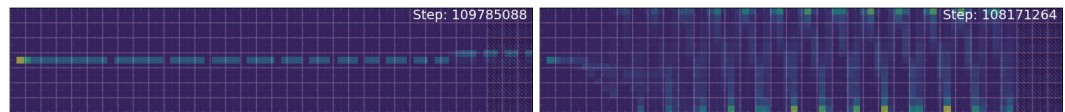

(a) The PPO agent plunges straight down the precipice, quickly reaching the bottom and neglecting fall damage.
(b) The PPOPID agent has learned to follow the winding path down the cave with minimal fall damage.

Figure 17: Heatmaps of visited location frequency superimposed on the 2D map of `Precipice Plunge` after training 100M timesteps on Level 1.

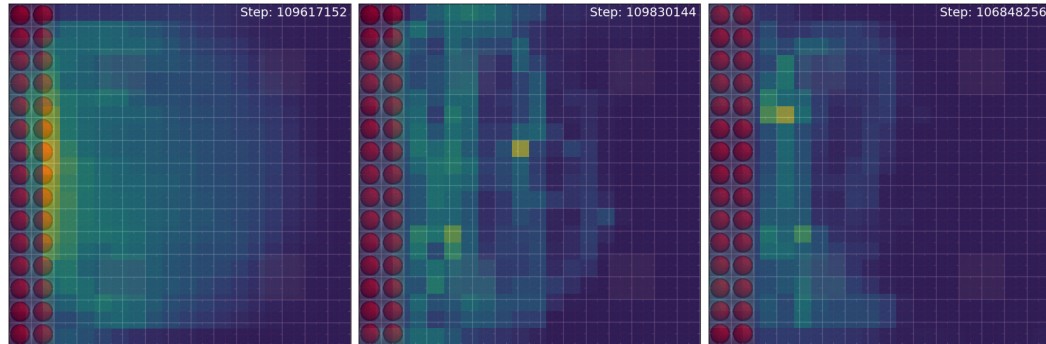

(a) **Level 1** has an almost flat surface, allowing the agent to navigate freely and cover most of the area for weapon delivery.
(b) **Level 2** features a more intricate terrain with obstacles, challenging the agent's ability to locate weapons and maneuver effectively.
(c) **Level 3** adds acid pits and decoy items that significantly constrain the agent, limiting its movement primarily to the starting area.

Figure 18: Heatmaps of visited location frequency superimposed on the 2D map of `Armament Burden` after training PPOPID for 100M timesteps.

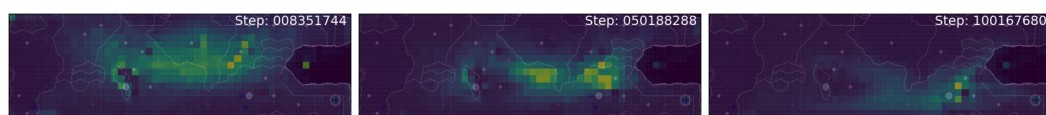

(a) Early in training, the agent concentrates its exploration around the center of the map.
(b) Midway through training, the focus shifts towards the lower region of the environment.
(c) By the end of training, the agent consistently favors the bottom-right area.

Figure 19: Heatmaps of visited location frequency during PPOPID training on Level 2 of `Detonator's Dilemma` over 100M timesteps, illustrating the agent's evolving spatial focus.

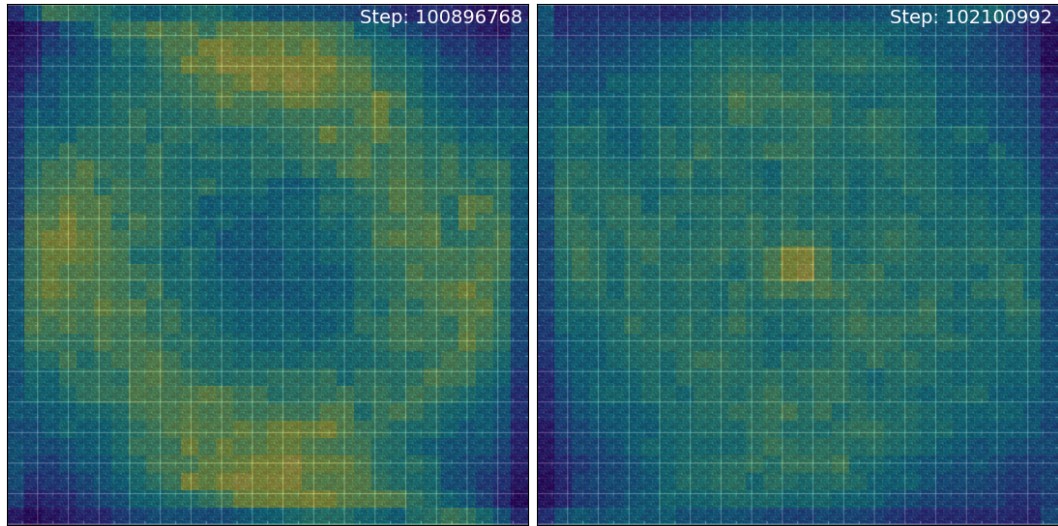

(a) The PPO agent repeatedly circles the environment to maximize item collection while neglecting lava.

(b) The PPOPID agent frequently traverses the central area to avoid lava.

Figure 20: Heatmaps of visited location frequency superimposed on the 2D map of `Volcanic Venture` after training 100M timesteps on Level 1.

## D  SIGNIFICANCE OF HASARD

HASARD's discrete simplified physics and pixelated graphics do not mirror the high-resolution imagery and complex dynamics required for autonomous driving or assistive robotics. Naturally, an agent excelling in HASARD would not be anywhere near capable when deployed in any of the aforementioned real-life scenarios. Nevertheless, there is substantial value in utilizing unrealistic simulation environments for foundational research in Safe RL. We list several key points motivating HASARD's design and utility:

**Computational Accessibility**   Incorporating highly realistic physics and visual rendering significantly increases the computational cost of simulations. HASARD is designed to be accessible for low-budget research settings, allowing more researchers to engage with vision-based Safe RL. Training RL algorithms in ultra-realistic environments with accurate physics and complex decision-making problems is not only costly but also time-intensive. By maintaining a balance between realism and computational demand, HASARD enables a tight feedback loop that facilitates rapid experimentation and iteration.

**Focus on Vision-Based Safety**   Much of recent Safe RL research has predominantly concentrated on continuous control problems, exemplified by the widely-used environments in Safety-Gymnasium (Ji et al., 2023a). HASARD aims to bridge the gap in safety considerations within vision-based learning, a domain with a wide range of applicability. The egocentric embodied perception in HASARD environments further introduces the problem of partial observability, which is often absent in prior works. The ultimate goal for many applications is to enable complex decision-making and precise manipulation under realistic physics, leveraging multimodal inputs of visual imagery and LIDAR. However, fulfilling all these criteria simultaneously poses a significant challenge for Safe RL research. To manage this, HASARD narrows the focus and decouples the problem, concentrating on visual perception of 3D environments, spatial awareness, and high-level control.

**Precedent in Research**   Video games, despite their lack of realism, have been instrumental in advancing AI research (Hu et al., 2024). They provide a controlled yet challenging environment for developing and testing new methodologies. Many game-based 3D benchmarks are adopted as useful tools within the community (Chevalier-Boisvert et al., 2024; Gong et al., 2023; Jeon et al., 2023), supporting the development of algorithms that can later be adapted to more realistic applications.

Similarly, simulation environments in Doom remain viable and relevant, as they have been widely used in recent research (Park et al., 2024; Kim et al., 2023; Zhai et al., 2023; Valevski et al., 2024).

**Solving Toy Problems**   Simulation environments with simplified physics and visuals can roughly emulate critical aspects of more complex systems. For example, much recent Safe RL research utilizes gridworld environments (Wachi et al., 2024a; Den Hengst et al., 2022), which, while basic, allow for the exploration of key concepts and strategies in safety and exploration. Similarly, HASARD, with its lower-resolution visuals and basic physics, still captures essential elements of navigating in a 3D space and introduces significant challenges related to egocentric Safe RL, such as depth perception, short-term prediction, and memory.

## E   COMPARISON WITH SAFETY-GYMNASIUM

Safety-Gymnasium is the most comprehensive and closely related benchmark to our work, making it an essential point of comparison. Although it consists of 28 environments, there are strong similarities between them. For instance, the six environments in the **Safe Velocity** suite all share the identical objective of the agent moving forward and are visually identical. These could effectively be considered a single environment with variations in the robot type.

Similarly, the four tasks in the **Safe Navigation** suite featuring different robots like `Point`, `Car`, `RaceCar`, `Doggo`, `Ant` share a single navigation objective and only slightly vary the objects within the environment. We argue that the variations within a single environment in HASARD offer equal or greater diversity than the differences observed among the `Goal`, `Button`, `Push`, `Circle` tasks in Safety-Gymnasium.

The **Safe Vision** suite introduces more visually diverse settings with environments such as `Building`, `Race`, `FormulaOne`. However, the core tasks remain focused on navigation, and higher difficulty levels simply introduce additional obstacles. Furthermore, these environments lack dynamic elements. The few entities in the `Building` task move in predictable, fixed patterns, and the environment remains static throughout the episode. The only changes observed are those directly caused by the agent.

In contrast, **HASARD** extends beyond mere navigation-based tasks, necessitating higher-order reasoning for task resolution. It incorporates randomly moving units, and exploits the third dimension more effectively, enabling entities to navigate vertical surfaces, resulting in more complex dynamics. Leveraging the ViZDoom game engine, HASARD allows for rapid environment simulation, while offering a broad and dynamic challenge that provides a comprehensive evaluation of many agent competencies, such as memory, short-term prediction, and distance perception.

### E.1   FRAMEWORK EFFICIENCY

To demonstrate the training efficiency of HASARD, we conducted a comparative analysis with Safety-Gymnasium (Ji et al., 2023a) on the same hardware. Among existing benchmarks tailored for Safe RL research, Safety-Gymnasium is arguably the most comprehensive suite and closest to our work, as it uniquely facilitates vision-based learning within 3D environments. We arbitrarily selected the SafetyPointGoal1-v0 task and ran the PPOLag implementation with default settings in Omnisafe (Ji et al., 2023b), a popular recent Safe RL library. We allowed two hours of training time for each framework. In Figure 21 and Table 13 we compare how many environment iterations is the framework able to facilitate and how frequent are the policy updates. We can see that HASARD outperforms Safety Gymnasium in both metrics by a large magnitude. Note, that HASARD runs with Sample-Factory *out-of-the-box*, whereas integrating Safety-Gymnasium with Sample-Factory would require substantial effort.

Table 13: Efficiency Comparison of Benchmarks

| Benchmark | Frames Per Second | Updates Per Second |
|---|---|---|
| Safety-Gymnasium | 180 | 0.03 |
| HASARD | 53985 | 15.12 |

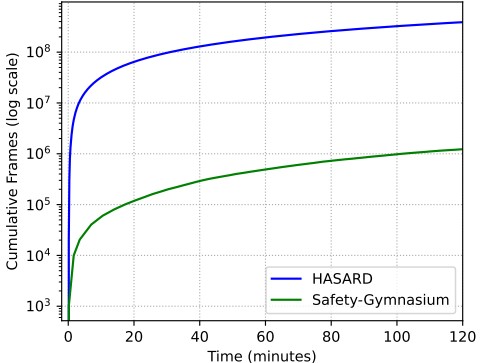 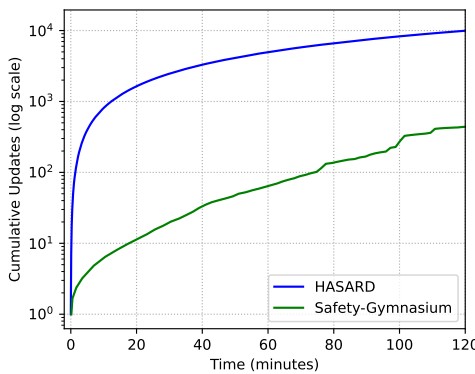

(a) HASARD is capable of accumulating frames at a rate several magnitudes greater than Safety-Gymnasium on the same hardware.

(b) The lack of effective parallelization results in a CPU rollout bottleneck due to which Safety-Gymnasium achieves far fewer policy updates.

Figure 21: Performance comparison of benchmarks over two hours of training on identical hardware. We trained a PPOLag agent on `VolcanicVenture1-v0` from HASARD and `SafetyPointGoal1-v0` from Safety-Gymnasium using default configurations.

Table 14: Human player rewards and costs averaged over 10 episodes compared to the PPOPID results with the simplified action space.

| Level | Method | Armament Burden R ↑ | C ↓ | Volcanic Venture R ↑ | C ↓ | Remedy Rush R ↑ | C ↓ | Collateral Damage R ↑ | C ↓ | Precipice Plunge R ↑ | C ↓ | Detonator's Dilemma R ↑ | C ↓ |
|---|---|---|---|---|---|---|---|---|---|---|---|---|---|
| 1 | PPOPID | 8.99 | 49.79 | 45.23 | 50.53 | 38.19 | 4.90 | 43.27 | 5.03 | 231.53 | 43.91 | 26.51 | 5.25 |
| | Human | 18.60 | 42.83 | 57.12 | 45.97 | 51.04 | 5.65 | 42.33 | 4.11 | 243.09 | 46.88 | 31.54 | 4.49 |
| 2 | PPOPID | 5.50 | 50.30 | 33.52 | 50.38 | 32.92 | 5.15 | 27.23 | 5.12 | 162.16 | 50.77 | 20.84 | 4.92 |
| | Human | 16.85 | 31.50 | 42.91 | 45.09 | 34.72 | 1.22 | 17.34 | 3.10 | 226.11 | 44.35 | 34.66 | 3.09 |
| 3 | PPOPID | 2.78 | 33.89 | 25.80 | 49.02 | 13.51 | 5.14 | 14.51 | 4.97 | 54.02 | 49.57 | 20.49 | 4.87 |
| | Human | 9.00 | 31.40 | 35.27 | 49.81 | 36.91 | 4.22 | 16.40 | 4.99 | 104.62 | 21.54 | 37.10 | 4.08 |

## F  THE HUMAN BASELINE

The question arises whether HASARD environments are already *solved*. Since the optimal strategy is unknown, we cannot establish an absolute performance ceiling. Although PPO, which ignores costs, might serve a reasonable upper bound for reward, our focus is on the highest reward achievable under safety constraints. To address this, we introduce a human baseline, similar to the approach in the Atari benchmark Mnih et al. (2013). We record the reward and cost of a human player averaged over 10 episodes per environment. Table 14 compares human performance with PPOPID, our highest-performing algorithm. Although the human player has access to the full action space of Doom, we compare the PPOPID results obtained with the simplified action space. As demonstrated in Appendix C.4 this setting yields far more favorable outcomes. The results indicate that PPOPID only achieves human-comparable performance on `Collateral Damage`, suggesting that other tasks are still far from being solved. It should be noted that the human baseline is likely to be suboptimal, implying ample room for improvement.

### F.1  WHAT MAKES HASARD COMPLEX?

In many instances at higher difficulty levels, the human player reached a higher score not due to more precise control, but through strategies that PPOPID and none of the other baselines managed to learn. After all, while the RL method can consistently output precise actions for each frame, the human is playing in real time at a *ticrate* of 35, leading to longer reaction times.

In `Armament Burden`, the agent has the option to discard all weapons. This strategy can be used to get rid of the very heavy weapons far from the starting zone, since carrying them back can be very costly. None of our evaluated baselines managed to learn this behavior. Instead, they simply avoided picking up heavy weapons altogether, as discarding weapons does not provide an immediate or obvious reward. Level 3 introduces decoy items that add to carrying load but do not grant any reward when delivered. None of the methods were able to figure out to avoid these items.

In Levels 2 and 3 of `Remedy Rush`, the environment becomes periodically dark after short time intervals, limiting the agent's ability to see the items it needs to collect or avoid. However, night vision goggles are located in the map, granting permanent vision when obtained. Despite this, none of the baselines developed a consistent strategy to seek out the goggles early in the episode before proceeding to with their item collection.

Level 3 of `Detonator's Dilemma` allows the agent to push the barrels to safer locations with few or no units nearby before detonating them. However, since pushing barrels does not yield immediate rewards, the baseline methods failed to adopt this strategy. Instead, the agents tend to avoid going near the barrels, as detonating them while standing too close often results in the agent harming itself and incurring a cost.

## G    EXTENDED FUTURE WORK

In this section, we will elaborate on several extensions to broaden the benchmark's application for other settings.

**Multi-Constraint**    In the current HASARD framework, each environment only has a single safety constraint. Given the focus on Safe RL, incorporating multiple safety constraints presents a compelling avenue. In the current version only `Detonator's Dilemma`, has two factors that increase the cost: (1) the agent harming neutral units and (2) the agent harming itself. Instead of combining them into one single value, we could decouple them into separate safety constraints. Similarly, a second constraint could be incorporated into `Armament Burden` for how frequently the agent is allowed to visit the delivery zone, and into `Volcanic Venture` for how often the agent can use the `JUMP` action.

**Multi-Agent**    The ViZDoom platform supports synchronous operations among multiple agents, allowing HASARD to be extended to collaborative MARL scenarios. `Precipice Plunge` does not provide any meaningful ways of collaboration, but in other environments multiple agents could collaborate by dividing the workload of the task, focusing on separate areas of the map.

**Transfer/Continual/Multi-task Learning**    The full action space setting is unified across all HASARD environments, allowing agents to use a consistent policy across all tasks. Moreover, there are many common elements, such as spatial navigation, entity behaviour, and environment dynamics. This paves the way for training a versatile agent capable of mastering all six tasks across three difficulty levels. This can be done in a multi-task or continual learning setting. A further interesting area of exploration is to what extent learned competencies are transferable across environments. For instance, the ability to successfully navigate an area without colliding with walls could be shared across tasks.

## H    TRAINING CURVES

Figure 23 depicts the training curves for **Level 1**. PPO maximizes the reward irrespective of associated costs, setting an upper bound for reward and cost in all environments with its unconstrained behavior. The primal-dual Lagrangian PPOPID and the penalty-based P3O manage to obtain the highest rewards while adhering to the default cost budget. PPOCost treats costs as negative rewards, occasionally yielding reasonable outcomes but without any guarantee of adhering to safety constraints. PPOLag closely meets the safety thresholds with some fluctuations, yet frequently yields lower rewards. PPOSauté has varied performance across tasks, often failing to satisfy the cost threshold.

In most tasks of **Level 2**, PPO also sets the upper bounds for reward and cost, as seen in Figure 24. PPOLag is the only baseline that consistently maintains the accumulated cost near the safety bud-

get. Conversely, PPOSauté struggles to lower its cost and stay within the safety budget in several environments.

As shown in Figure 25, PPO consistently dominates with high rewards accompanied by high costs also in **Level 3**. PPOLag and PPOSauté are unable to improve rewards while effectively controlling the cost in `Remedy Rush`, `Collateral Damage`, and `Precipice Plunge`, as the curves remain relatively flat. `Remedy Rush` demonstrates the fluctuating behavior of PPOLag originating from the Lagrangian optimization. Notably, PPOCost manages to obtain equal rewards to PPO in `Detonator's Dilemma` while maintaining a lower cost.

Figure 22 shows the training curves for the **hard constraint** setting of level 1. **PPO** maintains high returns due to a lack of an explicit safety feedback mechanism. In contrast, **PPOCost** and **PPOLag** exhibit overly conservative behavior by consistently selecting the passive `NO-OP` action, failing to learn a policy that achieves noticeable rewards while strictly adhering to safety constraints. Note that the cost threshold is $\xi = 0$ under hard constraints.

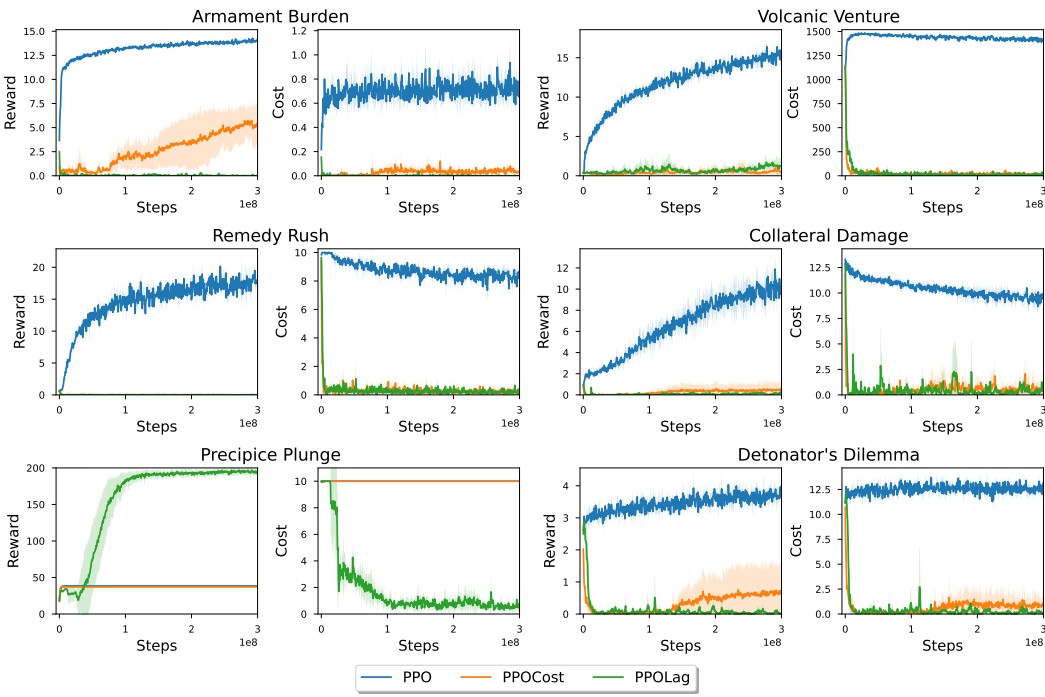

Figure 22: Training curves of HASARD Level 1 under **hard constraints** with 95% confidence intervals across five seeds.

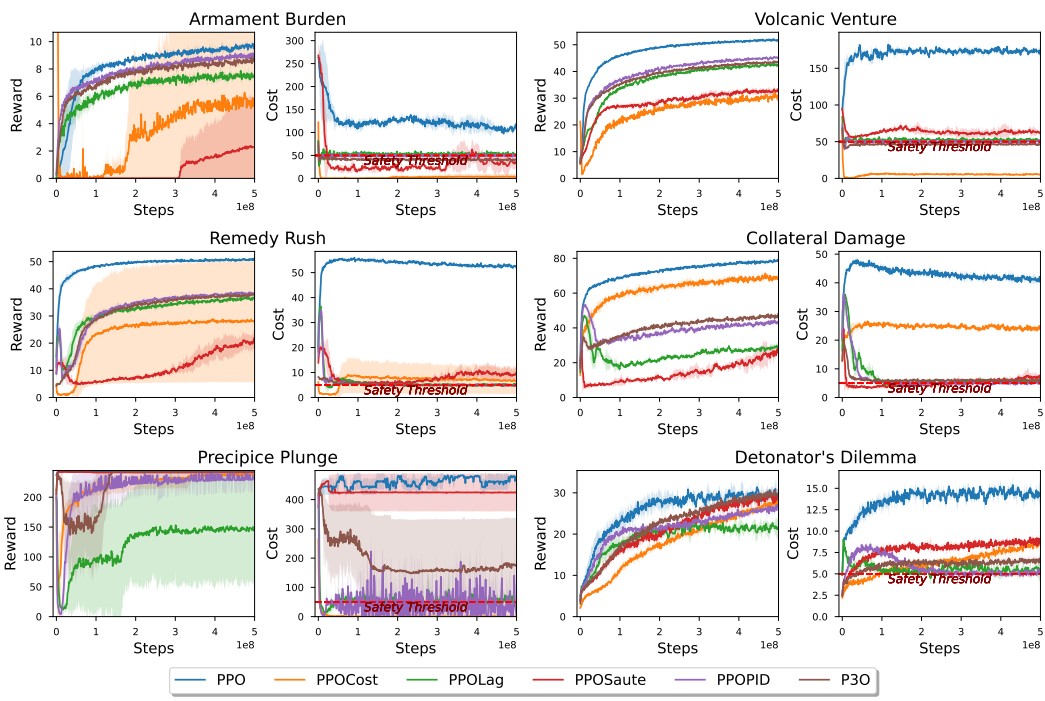

Figure 23: Training curves of HASARD **Level 1** with 95% confidence intervals across five seeds.

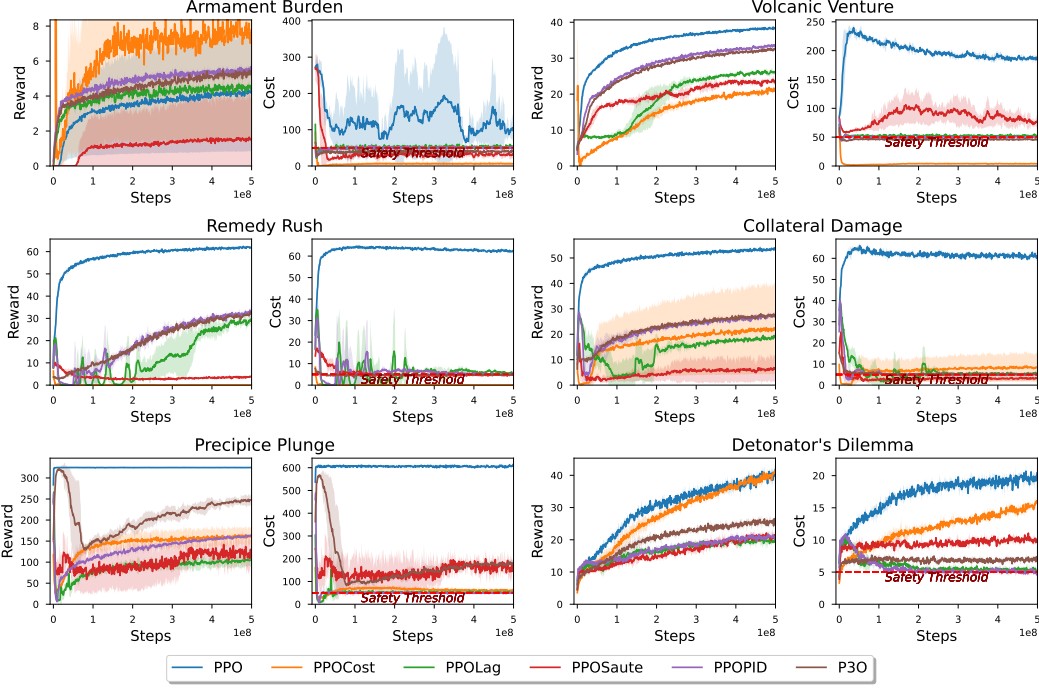

Figure 24: Training curves of HASARD **Level 2** with 95% confidence intervals across five seeds.

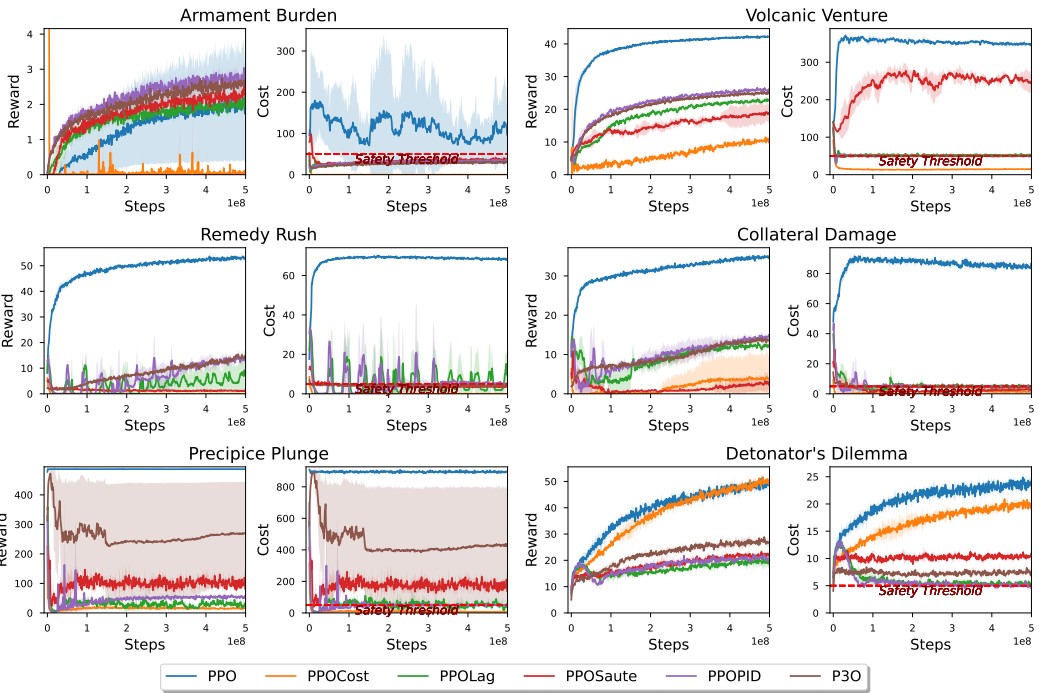

Figure 25: Training curves of HASARD **Level 3** with 95% confidence intervals across five seeds.

