# OpenReview forum: "HASARD: A Benchmark for Vision-Based Safe Reinforcement Learning in Embodied Agents"
_ICLR.cc/2025/Conference — ICLR 2025 Poster_

### Official Review · Reviewer_NtcD · 2024-10-21

**Soundness:** 3
**Presentation:** 3
**Contribution:** 3
**Rating:** 5
**Confidence:** 3

**Summary:**

This paper introduces the HASARD, a benchmark designed for egocentric pixel-based safe RL in diverse and stochastic 3D environments. Unlike existing benchmarks, HASARD emphasizes spatial comprehension, short-term planning, and active prediction for high rewards while ensuring safety. It offers three difficulty levels, supporting both soft and hard safety constraints. The benchmark includes heatmaps for visual analysis, aiding in strategy development. By targeting vision-based embodied safe RL, HASARD addresses the need for benchmarks mirroring real-world complexities. The paper's contributions include the design of six novel ViZDoom environments with safety constraints, integration with Sample-Factory for rapid simulation and training. Evaluation of baseline methods within HASARD highlights challenges in balancing performance and safety under constraints.

**Strengths:**

The paper demonstrates several notable strengths across the dimensions of originality, quality, clarity, and significance:



1. **Originality**: It introduces HASARD, a benchmark specifically designed for vision-based embodied safe reinforcement learning (RL) in complex 3D environments.



2. **Quality**: Comprehensive design of 6 diverse environments with 3 difficulty levels each, offering a range of challenges.



3. **Clarity**: The paper is structured in a logical and coherent manner, facilitating the understanding of complex concepts.



4. **Significance**: The paper Addresses an important need in safe RL research for more realistic and challenging benchmarks. It enables systematic evaluation and comparison of safe RL algorithms in vision-based 3D settings.

**Weaknesses:**

While the paper makes valuable contributions, several areas could be improved:



1. The paper refers to ViZDoom as a 3D environment, but its pixelated, less detailed graphics compared to modern 3D games challenge this characterization.



2. **Narrow Range of Baselines**: Evaluations focus primarily on PPO-based algorithms. Incorporating approaches like model-based safe RL or constrained policy optimization (e.g., https://arxiv.org/abs/2210.07573) would enhance the assessment.



3. **Limited Visual Input Analysis**: Though vision-based learning is emphasized, the paper lacks analysis of how visual complexity influences performance. Exploring different visual conditions (lighting, distractors) and comparing raw pixels with simplified representations would highlight the unique challenges of vision-based safe RL, especially since the visual inputs in the environment appear less realistic.



4. **Action Space Limitation**: Only discrete action spaces are supported. It is unclear how continuous safe RL algorithms would be benchmarked.



5. **Real-World Relevance**: The connection between the benchmark tasks and real-world safe RL challenges needs clearer articulation. Providing examples of practical applications would strengthen motivation.



Addressing these points could strengthen the paper and increase the impact and utility of the HASARD benchmark for the safe RL research community.

**Questions:**

1. Is ViZDoom truly a 3D environment, considering its graphics appear pixelated and less detailed compared to modern 3D games?



2. Why are the baseline algorithms limited to PPO-based approaches? Could the paper include more diverse methods, such as model-based safe RL or constrained policy optimization (e.g., https://arxiv.org/abs/2210.07573)?



3. How can continuous safe RL algorithms be benchmarked when the paper only supports discrete action spaces?

---

> ### Comment · Reviewer_NtcD · 2024-11-26
>
> I have not seen the author's response. I concur with the other reviewers that VisDoom lacks a detailed simulation of real-world physics.

---

> > ### Author Response · Authors · 2024-12-04
> >
> > We apologize for the delayed response. Adapting model-based methods like SafeDreamer to a new setting is a complex and time-intensive process due to their high sensitivity. We kindly request that the reviewer nevertheless take our responses into consideration.
> >
> > **W1 & Q1. ViZDoom is pixelated and not 3D**
> >
> > Indeed, ViZDoom uses rendering with sprites and precomputed lighting, whereas modern games use elements of true 3D graphics, such as dynamic lighting, shadows, and voxel-based rendering. However, this graphical complexity comes with a computational overhead.
> >
> > It is important to note that when benchmarking RL in simulation environments, the critical aspect is not necessarily how the environment is rendered under the hood, but how the environment appears to the agent. Accurate 3D dynamics are essential for contexts like robotics simulations and modern video games, where precise collision calculations and complex rigid-body interactions significantly impact the outcome or user experience. However, HASARD focuses on a different dimension of complexity.
> >
> > Our aim with HASARD is to expand the scope beyond basic navigation and continuous robotic motor control tasks. While environment suites like CARLA provide high-fidelity graphics and Safety-Gymnasium offers sophisticated physics simulations, HASARD serves as a complementary benchmark rather than a direct competitor. While realistic physics and lifelike environments are undoubtedly important, HASARD emphasizes fostering higher-order comprehension and decision-making.
> >
> > HASARD's scenarios require agents to perceive spatial relationships, accurately gauge depth, and predict the movement and future positions of entities. This focus on reasoning, planning, and decision-making allows HASARD to tackle a different dimension of safe RL. Additionally, HASARD achieves this at a fraction of the computational cost of physics-based simulation environments, making it particularly well-suited for sample-hungry, model-free approaches. For these purposes, HASARD effectively serves as a 3D benchmark. Although visually unrealistic, it closely replicates the intricacies of navigating the real world.
> >
> > **W2 & Q2. Narrow Range of Baselines**
> >
> > |Method|AB R↑|AB C↓|VV R↑|VV C↓|RR R↑|RR C↓|CD R↑|CD C↓|PP R↑|PP C↓|DD R↑|DD C↓|
> > |-----------|-------|-------|-------|-------|-------|-------|-------|-------|-------|-------|-------|-------|
> > |PPO|1.99|118.22|42.20|347.77|53.16|68.27|34.86|84.44|487.17|894.22|49.36|23.72|
> > |PPOCost|*0.04*|*0.05*|*10.61*|*14.76*|*0.01*|*0.02*|*3.93*|*1.35*|*15.39*|*6.64*|49.95|19.79|
> > |PPOLag|2.03|*31.89*|22.77|52.54|8.02|9.74|12.22|5.29|23.87|*34.98*|*19.27*|5.26|
> > |PPOSauté|2.32|*37.68*|18.89|246.80|*1.17*|*3.50*|*2.74*|*2.76*|101.64|176.14|22.20|10.36|
> > |PPOPID|*2.78*|*33.89*|*25.80*|*49.02*|13.51|5.14|*14.51*|*4.97*|*54.02*|*49.57*|*20.49*|*4.87*|
> > |P3O|*2.61*|*29.94*|*24.93*|*49.84*|*14.29*|*4.19*|13.57|*4.12*|269.14|428.72|26.73|7.49|
> > |SafeDreamer|*3.11*|*48.48*|*33.07*|*49.60*|*19.81*|*4.79*|***21.57***|*4.77*|71.22|44.03|36.33|*4.60*|
> > |Human|***8.03***|*46.43*|***38.06***|*41.12*|***34.21***|*3.17*|19.43|*4.77*|***147.32***|*47.45*|***43.90***|*4.02*|
> >
> > We have included **SafeDreamer** [1], the most recent state-of-the-art Safe RL method in our evaluation. Similarly to other benchmarks, SafeDreamer surpasses other baselines, achieving the highest results across all environments. However, it only surpasses the human baseline on *Collateral Damage*, a task requiring high precision and accurate timing. It does yet not manage to develop clever strategies in *Armament Burden* (discarding heavy weapons and avoiding decoy items), *Precipice Plunge* (restarting the episode when a safe action can no longer be performed), and *Remedy Rush* (immediately seeking out the night vision goggles to gain permanent vision). Therefore, in these environments the human baseline, although weaker in control and accuracy, still outperforms SafeDreamer thanks to a better strategy and knowledge of the task. The results of evaluating SafeDreamer show that HASARD environments provide ample room for algorithmic improvement, not to mention when also utilizing the full action space with 864 discrete and 2 continuous actions, which could make control far more efficient.
> >
> > [1] Huang, Weidong, et al. "Safe dreamerv3: Safe reinforcement learning with world models." arXiv preprint arXiv:2307.07176 (2023).

---

> > ### Author Response · Authors · 2024-12-04
> >
> > **W3. Limited Visual Input Analysis**
> >
> > To demonstrate the visual complexity of HASARD, we leverage the privileged information about the agent's observations in the ViZDoom framework. We create simplified representations through two strategies: segmentation and inclusion of depth information. We hypothesize that simplified representations and auxiliary visual information improve performance.
> >
> > **Segmentation**. ViZDoom provides a labels buffer that assigns a ground truth label to each pixel in the observation, identifying the item, unit, wall, or surface it belongs to, thereby effectively segmenting the scene. We assign a predefined fixed color to each such unique object that can be encountered in our environments. This makes the observations consistent across episodes and environments. With the help of the labels buffer, we then map every pixel in the observation to its respective color.
> >
> > We train the PPOPID and P3O agents on the segmented observations of *Armament Burden*. This task is suitable due to its high number of different objects: 7 types of weapons, 4 types of decoy items, and 4 types of obstacles. Factoring in the agent, walls, and surfaces, the environment comprises a total of 18 unique labels. The results comparison in the table below indicates that **the simplified representations make the task easier**.
> >
> > |Algorithm|Input|Reward|Cost|
> > |---------|-----|------|----|
> > |PPOPID|Default Obs|4.88|49.97|
> > |PPOPID|Segmentation|5.92|49.22|
> > |P3O|Default Obs|4.38|39.66|
> > |P3O|Segmentation|5.73|38.16|
> >
> > **Depth Information**. ViZDoom provides a depth buffer that assigns a value between 0-255 to each pixel in the observation, representing its relative distance from the agent. A value of 255 (fully white) indicates the farthest points, while 0 (fully black) denotes the closest. Intermediate values reflect their distance, relative to the closest and farthest points. This feature allows the agent to directly perceive the proximity of walls, surfaces, and objects.
> >
> > We evaluate this approach in the *Precipice Plunge* task, where depth plays a critical role. The agent must accurately gauge distances to platforms below as it attempts to safely descend the cave. Instead of relying solely on the depth buffer, we concatenate its values with the original RGB observations. As indicated by the results below, **learning solely from default observations is more challenging** due to the complexity of accurate depth perception.
> >
> > |Algorithm|Input|Reward|Cost|
> > |---------|-----|------|----|
> > |PPOPID|Default Obs|129.60|50.83|
> > |PPOPID|Default Obs + Depth Buffer|173.15|50.13|
> > |P3O|Default Obs|187.14|121.27|
> > |P3O|Default Obs + Depth Buffer|224.56|102.60|
> >
> >
> > **W5. Real-World Relevance**
> >
> > The challenges in HASARD do not lie in controlling agents under complex physics or comprehending hyper-realistic visuals but rather in fostering perception, reasoning, and decision-making. Below, we present two examples to further motivate our environments.
> >
> > **Industrial Automation**. Autonomous robots for warehouse logistics navigate simple but dynamic environments where it is important to accurately perceive their surroundings and detect relevant objects.  Unlike humanoids, quadrupeds, or precision-focused assembly robots, these systems have simpler action spaces and fewer degrees of freedom. In these settings, robots are tasked with picking and placing items, requiring effective path planning to optimize their operations while adhering to their carrying load (*Armament Burden*). They should accurately detect and avoid obstacles such as shelves, inventory, human workers, other robots, and restricted or unnavigable areas (*Remedy Rush* and *Volcanic Venture*). The robots ought to anticipate the movement and paths of other dynamic entities to avoid collisions, a challenge closely aligned with HASARD’s focus on predicting future object locations (*Collateral Damage* and *Detonator's Dilemma*). Lastly, due to partial observability, it is beneficial to recall the locations of recently encountered items and entities that are still nearby but no longer within their present field of view (*Armament Burden*).
> >
> > **Autonomous Drones**. Although the agent in HASARD cannot fly, the emphasis on accurately perceiving depth is directly applicable to the safe navigation of autonomous drones. Similar to how the agent in *Precipice Plunge* is tasked to vertically navigate the environment, drones frequently perform vertical maneuvers and must adjust their speed based on the proximity of surfaces and obstacles. General-purpose drones are usually controlled by 6 degrees of freedom (3 translational and 3 rotational), corresponding to 6 continuous actions. This complexity is comparable to HASARD’s full action space, which includes two continuous and 864 discrete actions. Similar to HASARD’s tasks, drones must anticipate the movement of other entities, avoid collisions, optimize their flight paths, and take note of recent observations to navigate effectively and safely.

---

> > ### Author Response · Authors · 2024-12-04
> >
> > **W4. Action Space Limitation & Q3. How to benchmark Continuous RL Methods?**
> >
> > We thank the reviewer for raising this point, as it is an important aspect of comparing HASARD with prior benchmarks. Indeed, HASARD does not support exclusively continuous action spaces, but discrete and hybrid (discrete + continuous). Real-world applications also often rely on discrete or hybrid action spaces, as seen in domains like healthcare [1], finance [2], robotics [3], agriculture [4], and energy systems [5]. Therefore, we would argue that it is rather a strength than a weakness that HASARD supports discrete and hybrid action spaces since prior popular Safe RL benchmarks (Safety-Gymnasium [6], Safe-Control-Gym [7], CARLA [8]) lack this setting. Only AI Safety Gridworlds [9] is uniquely made for discrete actions, but it is 2D and very simplistic.
> >
> > However, if a method is only compatible with continuous action spaces, then it
> > 1) reflects a limitation of the method's generality.
> > 2) means that it is tailored for a narrower use case. Similarly, some popular RL methods are specifically designed for discrete action spaces, such as MuZero [10], Agent57 [11], and Rainbow [12].
> > 3) could be adapted to support a discrete/hybrid action space with little ease and no noticeable performance decrease. Adapting a continuous-action method to support discrete action spaces is typically more straightforward than the reverse. This often involves modifying the action sampling process to output probabilities for each discrete action and using techniques like softmax for sampling during training. For instance, one only needs to replace the policy's Gaussian output distribution with a categorical one, such as for SAC-Discrete [13].
> >
> >
> > [1] Raghu, Aniruddh, et al. "Deep reinforcement learning for sepsis treatment." arXiv preprint arXiv:1711.09602 (2017).
> >
> > [2] Hambly, Ben, Renyuan Xu, and Huining Yang. "Recent advances in reinforcement learning in finance." Mathematical Finance 33.3 (2023): 437-503.
> >
> > [3] Zeng, Andy, et al. "Learning synergies between pushing and grasping with self-supervised deep reinforcement learning." 2018 IEEE/RSJ International Conference on Intelligent Robots and Systems (IROS). IEEE, 2018.
> >
> > [4] Abioye, Emmanuel Abiodun, et al. "Precision irrigation management using machine learning and digital farming solutions." AgriEngineering 4.1 (2022): 70-103.
> >
> > [5] Glavic, Mevludin, Raphaël Fonteneau, and Damien Ernst. "Reinforcement learning for electric power system decision and control: Past considerations and perspectives." IFAC-PapersOnLine 50.1 (2017): 6918-6927.
> >
> > [6] Ji, Jiaming, et al. "Safety gymnasium: A unified safe reinforcement learning benchmark." Advances in Neural Information Processing Systems 36 (2023).
> >
> > [7] Yuan, Zhaocong, et al. "Safe-control-gym: A unified benchmark suite for safe learning-based control and reinforcement learning in robotics." IEEE Robotics and Automation Letters 7.4 (2022): 11142-11149.
> >
> > [8] Dosovitskiy, Alexey, et al. "CARLA: An open urban driving simulator." Conference on robot learning. PMLR, 2017.
> >
> > [9] Leike, Jan, et al. "AI safety gridworlds." arXiv preprint arXiv:1711.09883 (2017).
> >
> > [10] Schrittwieser, Julian, et al. "Mastering atari, go, chess and shogi by planning with a learned model." Nature 588.7839 (2020): 604-609.
> >
> > [11] Badia, Adrià Puigdomènech, et al. "Agent57: Outperforming the atari human benchmark." International conference on machine learning. PMLR, 2020.
> >
> > [12] Hessel, Matteo, et al. "Rainbow: Combining improvements in deep reinforcement learning." Proceedings of the AAAI conference on artificial intelligence. Vol. 32. No. 1. 2018.
> >
> > [13] Christodoulou, Petros. "Soft actor-critic for discrete action settings." arXiv preprint arXiv:1910.07207 (2019).

---

### Official Review · Reviewer_Rd3K · 2024-11-03

**Soundness:** 2
**Presentation:** 2
**Contribution:** 2
**Rating:** 5
**Confidence:** 4

**Summary:**

The paper proposes a new egocentric vision-based 3D simulated environment for benchmarking safe reinforcement learning. The benchmark is more realistic and challenging compared to common prior safe RL benchmark environments. In addition, the paper has evaluations for some safe RL algorithms on the proposed benchmark demonstrating its feasibility of use and the potential for building better approaches to perform more favorably on it.

**Strengths:**

- The paper is well motivated and targets an important problem - that of building realistic and reliable RL benchmarks, and more specifically benchmarks for safe RL. This involves addressing challenges with the simple natural of prior benchmarks - both visually and in terms of higher dimensional action space and increased temporal horizons.

- The proposed benchmark HASARD is built on top of an existing game engine VizDoom and is able to inherit all of its properties for re-use. The multiple levels in HASARD can be potentially helpful in evaluating different notions of safety in proposed safe RL algorithms.

- The paper has detailed evaluations of several safe RL algorithms on HASARD indicating that the framework is feasible for training constrained RL policies. The evaluations reveal that simple algorithms based on PPO and constrained PPO can achieve non-trivial performance in the benchmark and also reasonable constraint satisfaction. It is good to see that these simple algorithms do not saturate the benchmark and there is still a lot of room for improvement.

**Weaknesses:**

- Unfortunately, while the paper is a decent attempt at building a safe RL benchmark, I am not convinced the safe RL community will be incentivized to use it. The main reason is that the notions of constraints in this benchmark are not directly tied to the very pragmatic safety considerations that need to be tackled in the real world - ranging from control systems to robotic deployments.

- The benchmark feels a bit incremental compared to the already existing VizDoom framework that has been around for years. The modifications for the different levels and environments in this framework do not capture the notions of open-world generalization and realism the field is headed towards in terms of evaluating RL systems. In addition, a lot of prior safe RL works have bechmakred their systems on real-world systems like robotic navigation and manipulation, and I am not convinced that a modified VizDoom framework is likely to create a reasonable impact in the community.

- The evaluations are all with variants of PPO and no other safe RL algorithms are tested. It is unclear why this is the case, since in my understanding the benchmark should not be tied to a particular type of algorithm

**Questions:**

Please refer to the weaknesses above:

- Unfortunately, while the paper is a decent attempt at building a safe RL benchmark, I am not convinced the safe RL community will be incentivized to use it. The main reason is that the notions of constraints in this benchmark are not directly tied to the very pragmatic safety considerations that need to be tackled in the real world - ranging from control systems to robotic deployments. Could the authors clarify how exactly they envision this benchmark to drive innovation in the safe RL community? And what sub-field of researchers would be likely to use it?

- The benchmark feels a bit incremental compared to the already existing VizDoom framework that has been around for years. Can the authors clarify if the proposed modifications are non-trivial and if they can be broadly applied to potentially other frameworks like Minecraft and other games?

- The evaluations are all with variants of PPO and no other safe RL algorithms are tested. It is unclear why this is the case, since in my understanding the benchmark should not be tied to a particular type of algorithm. Please clarify the evaluations and if there is any specific assumption on the type of safe RL algorithms that could be tested on the benchmark?

- Can the authors make 1-1 comparisons with the proposed benchmark and the features of prior simulated and real world benchmarks that have been used by safe RL papers in  the past?

---

> ### Author Response · Authors · 2024-11-20
>
> **W1 & Q1. Unpragmatic Constraints**
>
> We acknowledge the significance of safety and accurate motor control in robotics and control systems, which has been a central focus within the safe RL community. However, as the scope of safety in RL extends to broader domains beyond complex robotic motor control, including autonomous navigation [1], healthcare [2], LM safety [3], and energy systems management [4], there is a need for safe RL methods that could generalize across this broad domain spectrum. Such methods should be capable of addressing a wide range of safety constraints, thereby expanding the applicability and impact of safe RL to yet unexplored domains.
>
> To support the development of such methods, we require different environments and benchmarks that facilitate this, rather than all of them narrowly tailoring to individual, pragmatic applications. Relying solely on robotic simulation suites like Safety-Gymnasium as the gold standard narrows the scope. The community could benefit from embracing a broader spectrum of meaningful environments that challenge diverse competencies and encourage generalizable solutions.
>
> HASARD does not directly address any of these specific application domains, nor does it attempt to cover the entire range of safety aspects, but it does extend beyond conventional 2D environments and widely used robotic control tasks. Instead of focusing on realistic physics or lifelike simulations, HASARD emphasizes fostering decision-making in visually complex and challenging settings, offering a fresh perspective for advancing the field.
>
> Not all constraints in HASARD are entirely disconnected from real-world settings. Although ViZDoom does not facilitate the accurate motor control required to match the complexity of the real world, it simulates the challenges of decision-making in visually intricate environments, such as perceiving depth.
>
> In *Precipice Plunge*, the agent has to determine whether stepping or jumping down to a lower platform is safe by assessing whether it's close enough to avoid fall damage. This parallels the challenges faced by robots navigating complex terrains.
>
> In *Armament Burden*, the agent must learn how each item contributes to its carrying load and balance this against how much weight it can carry. It should plan a good path to maximize its time efficiency. It must also factor in the distance to the delivery zone, evaluating whether the potential reward for delivering an item justifies temporarily exceeding its carrying capacity. These characteristics could be tied to logistics or supply chain optimization, such as where a robot needs to manage the warehouse inventory.
>
> In *Collateral Damage* and *Detonator's Dilemma*, the agent must anticipate the future positions of other entities by analyzing their speed and past trajectories. This mirrors a similar competency in autonomous driving, where safety hinges on accurately predicting the movements of nearby vehicles and pedestrians.
>
> The focus of safety here is less about precise control problems and more about strategic decision-making. While we acknowledge that these game-based environments are detached from reality, they effectively simulate high-level challenges and complexities. Recent studies in safe RL continue to utilize simplified, unrealistic environments for experimental evaluation, such as grid worlds [5, 6, 7], to investigate some very intricate aspects of their method.  These simplified settings remain relevant due to their interpretability, which is often lacking in more realistic environments. Additionally, they are not directly tied to pragmatic safety constraints. In contrast, ultra-realistic and highly complex environments make it challenging to isolate specific factors.
>
> [1] Nehme, Ghadi, and Tejas Y. Deo. "Safe Navigation: Training Autonomous Vehicles using Deep Reinforcement Learning in CARLA." arXiv preprint arXiv:2311.10735 (2023).
>
> [2] Cao, Junyu, Esmaeil Keyvanshokooh, and Tian Liu. "Safe reinforcement learning with contextual information: Theory and application to personalized comorbidity management." Available at SSRN 4583667 (2023).
>
> [3] Wachi, Akifumi, et al. "Stepwise alignment for constrained language model policy optimization." arXiv preprint arXiv:2404.11049 (2024).
>
> [4] Huo, Xiang, et al. "Optimal Management of Grid-Interactive Efficient Buildings via Safe Reinforcement Learning." arXiv preprint arXiv:2409.08132 (2024).
>
> [5] Garcıa, Javier, and Fernando Fernández. "A comprehensive survey on safe reinforcement learning." Journal of Machine Learning Research 16.1 (2015): 1437-1480.
>
> [6] Wachi, Akifumi, et al. "Safe exploration in reinforcement learning: A generalized formulation and algorithms." Advances in Neural Information Processing Systems 36 (2024).
>
> [7] Den Hengst, Floris, et al. "Planning for potential: efficient safe reinforcement learning." Machine Learning 111.6 (2022): 2255-2274.

---

> ### Author Response · Authors · 2024-11-21
>
> **W2 & Q2. Incremental to ViZDoom.**
>
> While HASARD builds upon the ViZDoom platform, it is far from a mere incremental extension of the original ViZDoom environments. Not only do they lack the means of evaluating safe RL, but are static and overly simplistic in terms of variety, visuals, map layout, and objectives, focusing on simple navigation to a target location (*my_way_home*, *deadly_corridor*, *health_gathering*) or turning/sidestepping and shooting enemies (*defend_the_center*, *defend_the_line*, *predict_position*). Additionally, all the original ViZDoom scenarios are restricted to flat, 2D surfaces, limiting their complexity and challenge. In contrast, we carefully designed the HASARD environments with more complex terrains and visuals, also introducing a notion of **cost** and exhibiting reward-cost trade-offs. We've incorporated novel **dynamic elements** in each environment: 1) changing and moving platforms in *Volcanic Venture*, 2) various items spawns and random pitfall locations in *Armament Burden*, 3) units with sporadic behaviour and random entity and object spawns in *Collateral Damage* and *Detonator's Dilemma*, 4) random layout and moving platforms in *Precipice Plunge*, and 5) dynamic lighting and item spawns in *Remedy Rush*. Additionally, HASARD introduces difficulty levels, which are absent in the original ViZDoom scenarios. The levels are not merely achieved by tweaking parameters but introduce novel elements.
>
> We agree with the reviewer that advancing towards open-world generalization and realism is crucial for the field. However, to our knowledge, the majority of recent works in the field continue to rely on platforms like Safety-Gymnasium [1, 2, 3, 4, 5], MetaDrive [3], Safe Control Gym [6], MuJoCo [5, 6], or RoboSuite [6], which are far removed from realistic applications. We would appreciate it if the reviewer could provide references to recent works that utilize real-world settings or highly realistic environments for Safe RL or benchmark their systems on real-world systems.
>
> Using ultra-realistic simulation environments for research poses challenges of computational overhead. Incorporating highly detailed physics and advanced visual rendering substantially increases the cost and time required for simulations. HASARD is intentionally designed to strike a balance between realism and computational efficiency, making it accessible to low-budget research settings. This approach broadens access to vision-based Safe RL research, enabling not only more researchers to engage with the field but also to expand the scope of evaluation from robotic motor control. Note, that ViZDoom still remains widely adopted in recent research [7, 8, 9, 10].
>
> [1] Zhao, Weiye, et al. "Guard: A safe reinforcement learning benchmark." arXiv preprint arXiv:2305.13681 (2023).
>
> [2] Hoang, Huy, Tien Mai, and Pradeep Varakantham. "Imitate the good and avoid the bad: An incremental approach to safe reinforcement learning." Proceedings of the AAAI Conference on Artificial Intelligence. Vol. 38. No. 11. 2024.
>
> [3] Huang, Weidong, et al. "Safe dreamerv3: Safe reinforcement learning with world models." arXiv preprint arXiv:2307.07176 (2023).
>
> [4] Zhao, Weiye, et al. "Implicit Safe Set Algorithm for Provably Safe Reinforcement Learning." arXiv preprint arXiv:2405.02754 (2024).
>
> [5] Li, Zeyang, et al. "Safe reinforcement learning with dual robustness." IEEE Transactions on Pattern Analysis and Machine Intelligence (2024).
>
> [6] Gu, Shangding, et al. "A Review of Safe Reinforcement Learning: Methods, Theories and Applications." IEEE Transactions on Pattern Analysis and Machine Intelligence (2024).
>
> [7] Park, Junseok, et al. "Unveiling the Significance of Toddler-Inspired Reward Transition in Goal-Oriented Reinforcement Learning." Proceedings of the AAAI Conference on Artificial Intelligence. Vol. 38. No. 1. 2024.
>
> [8] Kim, Seung Wook, et al. "Neuralfield-ldm: Scene generation with hierarchical latent diffusion models." Proceedings of the IEEE/CVF conference on computer vision and pattern recognition. 2023.
>
> [9] Zhai, Yunpeng, et al. "Stabilizing Visual Reinforcement Learning via Asymmetric Interactive Cooperation." Proceedings of the IEEE/CVF International Conference on Computer Vision. 2023.
>
> [10] Valevski, Dani, et al. "Diffusion models are real-time game engines." arXiv preprint arXiv:2408.14837 (2024).

---

> ### Author Response · Authors · 2024-11-21
>
> **W3 & Q3. Only PPO variations**
>
> It just so happens that the best versions of these popular safe RL methods work best with PPO. Most of the Safe RL algorithms we evaluated are not constrained to a specific base RL algorithm, and neither is the benchmark. The PPO variants are known to have the best performance. We also included TRPO implementations of some safe RL methods. We present the results of Level 1 in the following table. The results satisfying the safety constraint are depicted in *cursive*, and the one of these with the highest reward is depicted in **bold**.
>
> We acknowledge the concern regarding the focus on PPO variants in our evaluations and would like to clarify that the benchmark itself is not tied to any specific type of algorithm. Equally, the Safe RL methods we selected are inherently independent of any particular base RL algorithm. However, in the original papers, the authors frequently utilized PPO as a base for their model-free methods. We therefore only included the PPO-based version not only because those have achieved higher results on prior benchmarks, but also not to compare apples with oranges, i.e., not to conflate differences in performance due to the choice of the base RL algorithm.
>
> To provide additional context, we also included TRPO-based implementations for some Safe RL methods. In the following table, we present the results for Level 1. Results that meet the safety constraint are denoted in *italic*, and the highest reward safe result is depicted in **bold**.
>
>
> |Method|AB R↑|AB C↓|VV R↑|VV C↓|RR R↑|RR C↓|CD R↑|CD C↓|PP R↑|PP C↓|DD R↑|DD C↓|
> |------|-----|-----|-----|-----|-----|-----|-----|-----|-----|-----|-----|-----|
> |PPO|9.68|109.30|51.64|172.93|50.78|52.44|78.61|41.06|243.42|475.62|29.67|14.28|
> |TRPO|7.24|144.79|41.00|185.30|45.70|48.47|67.91|37.29|243.41|468.99|20.03|8.92|
> |PPOLag|7.51|52.41|42.40|52.00|36.37|5.25|29.09|5.61|*147.24*|*44.96*|21.49|5.62|
> |TRPOLag|*4.06*|*48.61*|***30.48***|*49.91*|*21.69*|*4.98*|31.63|7.28|177.93|309.16|16.84|6.85|
> |PPOPID|**8.99**|*49.79*|45.23|50.53|***38.19***|*4.90*|**43.27**|5.03|***231.53***|*43.91*|**26.51**|5.25|
> |TRPOPID|*0.36*|*8.18*|23.81|50.93|*10.36*|*4.54*|34.74|10.13|172.87|299.46|12.81|6.84|
>
> Similar to prior findings, the TRPO versions of these methods consistently perform noticeably worse than their PPO counterparts. Also note, that evaluating all possible Safe RL methods is beyond the scope of our work.
>
> **Q4. 1-1 Comparisons with Prior Benchmarks?**
>
> Could the reviewer clarify what specific features or real-world benchmarks they are referring to? It is not entirely clear to us what exactly is being requested. We've included a performance comparison with Safety-Gymnasium in **Appendix C** of the paper. Among prior Safe RL benchmarks, Safety-Gymnasium stands out as both the most widely adopted in recent research and the most complex and similar to our work. Please let us know what other features or benchmarks we should compare.

---

> > ### Comment · Reviewer_Rd3K · 2024-11-24
> > **Response to rebuttal**
> >
> > Dear authors,
> >
> > Thank you for attempting to address the comments from all the reviewers. Unfortunately, I am still not convinced about the significance of the proposed benchmark, the applicability of algorithms tested on it to other more realistic scenarios where safety would be actually meaningful, and I agree with comments by other reviewers regarding lack of physical realism in VizDoom and the lack of relevant modern baselines.
> >
> > As such, I am not able to support recommend accepting the paper.

---

> > > ### Author Response · Authors · 2024-12-04
> > >
> > > Thank you for reviewing our response. We have made further efforts to address the concerns raised by all reviewers:
> > >
> > > **Inclusion of Modern Baselines**. We integrated the state-of-the-art SafeDreamer baseline to demonstrate the relevance and complexity of HASARD. This inclusion further highlights that the benchmark remains unsolved.
> > >
> > > **Real-World Applicability**. In our response to Reviewer NtcD, we provided examples of realistic applications where the HASARD environments could be highly relevant.
> > >
> > > **Physical Realism**. We acknowledge the lack of physical realism in our chosen platform and have thoroughly addressed this point in our responses, as it was raised by multiple reviewers.
> > >
> > > We kindly ask the reviewer to consider our recent responses to these issues provided to other reviewers, as they directly address the core concerns outlined in your feedback. We appreciate your time and evaluation of our work.

---

### Official Review · Reviewer_NMGZ · 2024-11-04

**Soundness:** 2
**Presentation:** 3
**Contribution:** 2
**Rating:** 5
**Confidence:** 4

**Summary:**

HASARD is a benchmark testing platform specifically designed for safe reinforcement learning, based on ViZDoom, providing a diverse range of 3D environments.

1. The tasks on this platform require agents to pursue high rewards while considering safety strategies, moving beyond simple 2D navigation to incorporate complex elements such as spatial understanding.
2. HASARD offers three difficulty levels and supports both soft and hard safety constraints, flexibly adapting to varying safety requirements.
3. The platform integrates Sample-Factory, enabling high-speed simulation that allows agents to address real-world safety challenges while reducing computational costs.
4. HASARD includes six environments based on ViZDoom and benchmarks various methods to demonstrate the limitations of existing technologies.

**Strengths:**

1. The authors tested six baseline algorithms on HASARD and provided an analysis of the results.
2. The tasks move beyond simple 2D navigation to incorporate complex elements such as spatial understanding

**Weaknesses:**

1. The reviewer believes that if the distinction between soft and hard constraints is merely based on whether the threshold is $0$, then other benchmarks share this characteristic, making this claim somewhat unsubstantiated.
2. Although multiple methods were tested in the current experiments, there is a lack of analysis on performance under different safety budgets. It is recommended to include experiments with varying safety thresholds to better understand the trade-off between safety and reward for each algorithm.
3. HASARD is based on the ViZDoom game engine, which, while computationally inexpensive, lacks detailed simulation of real-world physics.
4. The anonymous video link provided by the authors is inaccessible.

**Questions:**

1. The article does not provide an in-depth analysis of performance under different safety budgets. Is there a plan to supplement the experiments with varying safety thresholds to comprehensively demonstrate the trade-offs between reward and safety for each algorithm? This would be very helpful in understanding the adaptability of different methods under various safety requirements.
2. Considering the limitations of ViZDoom in simulating real-world physics, have the authors explored other engines with superior physical simulation capabilities (e.g., Isaac Gym)?

**Details Of Ethics Concerns:**

No ethics concerns.

---

> ### Author Response · Authors · 2024-11-14
>
> **W1. Hard Constraints**
>
> This concern is well-founded. We wish to clarify that our introduction of hard constraints is not presented as a novel contribution, nor is it central to our paper, as similar setups can indeed be implemented in other libraries. However, it is worth noting that simply decreasing the cost threshold to 0 is not our only modification. Additional task-specific adjustments have been made; for example, harsher penalties apply upon incurring costs, often terminating the episode abruptly. In *Armament Burden*, for instance, the agent loses all acquired weapons if it exceeds carrying capacity.
>
> Our primary focus here lies in the evaluation protocol introduced by HASARD, which emphasizes the integration and assessment of safety constraints in RL training. HASARD supports evaluations across diverse safety formulations [1], enabling fairer comparisons by aligning the safety problems that different algorithms are designed to address. If the reviewer yet nevertheless feels that the emphasis on hard constraints seems overstated or the claims appear unsubstantiated, we are open to adjust the writing to reflect this.
>
>
> **W2 & Q1. Different Safety Budgets**
>
> Understanding the trade-off between safety and reward is indeed crucial for evaluating how algorithms adapt to varying safety requirements. We addressed this in Appendix B.2 (referenced in Section 5.1), where we examined the effects of different cost budgets by testing PPOLag and PPOSauté on Level 1 of HASARD, with both higher and lower safety thresholds than the original setting. The results in Figure 14 nicely show that stricter safety bounds lead to lower rewards, with *PPOSauté* exhibiting greater sensitivity than *PPOLag*. We are open to incorporating additional safety thresholds and other baselines if the reviewer believes this would strengthen our work.
>
> If the reviewer feels further details on these results would strengthen the main text, we are open to incorporating the key findings on these trade-offs directly into the main paper. This could potentially replace the hard constraints results section, which may have been overstated (as discussed above). We welcome and would appreciate the reviewer’s feedback on this matter.

---

> ### Author Response · Authors · 2024-11-14
>
> **W3 & Q2. ViZDoom Lacks Realism**
>
> While we acknowledge that engines like Isaac Gym offer advanced physics simulation, our aim is to expand the scope not only beyond conventional 2D environments (listed as a strength by the reviewer), but also continuous control robotic manipulation tasks. Rather than focusing on realistic physics or lifelike environments, our emphasis is on fostering decision-making in visually intricate settings. Notably, many recent studies in safe RL continue to utilize simplified, unrealistic environments for experimental evaluation [2, 3], indicating their ongoing relevance. ViZDoom, in particular, remains widely adopted in recent research [4, 5, 6, 7]. Appendix F of the paper provides further rationale for our choice of ViZDoom.
>
> Figure 18b in Appendix G depicts the learning curves for Level 3 tasks. Training has not fully converged across multiple methods even after 5e8 environment iterations, underscoring the complexity of these environments and the high iteration count needed to approach peak performance. Despite the complexity of the high-dimensional pixel-based inputs, the framework was able to simulate 500M frames and perform 150K policy updates in a training time of ~3 hours on a single GPU. This complexity doesn’t stem from advanced physics or intricate motor control but instead from demands like precise navigation, spatial awareness, depth perception, and predicting the movement of other entities. Combining these challenges with accurate physics and continuous motor control would make the problem unfeasible to solve within a reasonable timeframe on standard hardware. Built on top of ViZDoom, HASARD therefore offers a balanced approach, blending fast simulation with complex tasks that remain meaningfully challenging.
>
>
> **W4. Inaccessible Video**
>
> We tried accessing the video hosted on [emalm](https://emalm.com/?v=dGLYX) from multiple devices and did not encounter any issues. To ensure the reviewer can view the demo, we have also re-uploaded the video to the [jumpshare](https://jumpshare.com/v/abSD9rLLXQDdkIXkQ3c7) platform. Please let us know if further adjustments are needed.
>
> We thank the reviewer for their feedback! Please let us know if there is anything else we ought to address.
>
> [1] Wachi, Akifumi, Xun Shen, and Yanan Sui. "A Survey of Constraint Formulations in Safe Reinforcement Learning." arXiv preprint arXiv:2402.02025 (2024).
>
> [2] Wachi, Akifumi, et al. "Safe exploration in reinforcement learning: A generalized formulation and algorithms." Advances in Neural Information Processing Systems 36 (2024).
>
> [3] Den Hengst, Floris, et al. "Planning for potential: efficient safe reinforcement learning." Machine Learning 111.6 (2022): 2255-2274.
>
> [4] Park, Junseok, et al. "Unveiling the Significance of Toddler-Inspired Reward Transition in Goal-Oriented Reinforcement Learning." Proceedings of the AAAI Conference on Artificial Intelligence. Vol. 38. No. 1. 2024.
>
> [5] Kim, Seung Wook, et al. "Neuralfield-ldm: Scene generation with hierarchical latent diffusion models." Proceedings of the IEEE/CVF conference on computer vision and pattern recognition. 2023.
>
> [6] Zhai, Yunpeng, et al. "Stabilizing Visual Reinforcement Learning via Asymmetric Interactive Cooperation." Proceedings of the IEEE/CVF International Conference on Computer Vision. 2023.
>
> [7] Valevski, Dani, et al. "Diffusion models are real-time game engines." arXiv preprint arXiv:2408.14837 (2024).

---

> > ### Comment · Reviewer_NMGZ · 2024-11-25
> >
> > I am pleased to see that, compared to my previous review of this paper, the authors have provided more insightful motivations and perspectives in their rebuttal during the discussion with other reviewers regarding the task’s motivation and details. If these insights can be well-integrated into the paper along with more comprehensive experiments and analyses, it will significantly strengthen the contributions of this work.
> > Additionally, I appreciate the authors' strong defense of their starting point and acknowledge the validity of most of their arguments.
> > Besides, I noticed that the authors discussed with Reviewer FzXY the use of OmniSafe. Upon reviewing the authors’ publicly available code, I found a folder named "OmniSafe," and it seems that the P3O algorithm only appears within it. However, the authors stated that they used Sample-Factory, which leaves me confused.

---

> > > ### Author Response · Authors · 2024-11-25
> > >
> > > Thank you for your thoughtful review and for acknowledging the improvements in our rebuttal.
> > >
> > > Since OmniSafe is a well-established Safe RL library, we initially used it to validate baseline performance consistency with our Sample-Factory implementations. However, since Sample-Factory offers significantly faster training speeds, we used it for all the experiments in the paper. The implementation of P3O can be found in *sample_factory/algo/learning/p3o_learner.py*.
> > >
> > > As OmniSafe does not natively support discrete action spaces or pixel-based observations, we extended the library to handle these requirements. To provide flexibility for benchmark users who may prefer OmniSafe, we included our extended version as a local module in the repository rather than relying on the pip-installed version. Before making the codebase public, we will update the documentation to make this setup and purpose more transparent.
> > >
> > > If there are any further weaknesses in our work or points that need to be clarified, we are happy to address them. Thank you again for your feedback.

---

> > > > ### Comment · Reviewer_NMGZ · 2024-11-26
> > > >
> > > > The reviewer thanks the authors for their response. I now understand your point.

---

> > > > > ### Author Response · Authors · 2024-12-04
> > > > >
> > > > > We greatly appreciate your thoughtful engagement with our work. As we have not detected any further concerns from the reviewer, we kindly ask that you consider increasing the score if you believe we have successfully addressed all the points raised. We also hope the additional experiments and analyses conducted during the rebuttal period, in response to other reviewers' feedback, further strengthen the contributions of our work.

---

### Official Review · Reviewer_FzXY · 2024-11-04

**Soundness:** 3
**Presentation:** 3
**Contribution:** 3
**Rating:** 8
**Confidence:** 4

**Summary:**

The paper presents HASARD, a benchmark tailored to vision-based safe reinforcement learning (RL) using egocentric, pixel-based inputs. Built on the ViZDoom platform, HASARD comprises six unique 3D environments across three difficulty levels, each designed to test safe RL in increasingly complex and dynamic scenarios. The benchmark allows for a range of agent objectives, from navigation to item collection and hazard avoidance, focusing explicitly on embodied safe RL with vision-based inputs.

**Strengths:**

Diverse Scenarios: HASARD provides varied 3D environments with different objectives and challenges, such as item collection, navigating hazardous terrain, and avoiding neutral units. This variety enriches the learning and testing possibilities, ensuring that the benchmark assesses both task performance and safety considerations.

Structured Curriculum: By offering three difficulty levels, HASARD presents a built-in curriculum for training RL agents, allowing gradual learning in increasingly challenging conditions. This approach is effective for developing robust agents that can generalize to new, more complex scenarios.

**Weaknesses:**

Outdated Baselines: All the baseline algorithms were published over two years ago, and the original implementations of these baselines do not support visual inputs. The lack of SOTA vision-input baselines, such as Lambda [1], Safe SLAC [2], and SafeDreamer [3], limits the benchmark’s relevance in evaluating current state-of-the-art safe RL methods.

Solvability by Existing Algorithms: Have the tasks introduced in this framework already been solved by existing algorithms? For instance, can PPO-PID successfully address these tasks? Are there settings within HASARD that current algorithms struggle to handle? By not including experiments with the latest baselines, it is unclear whether the HASARD benchmark will drive the development of new algorithms or simply reaffirm existing solutions.

Task Complexity:
What is the primary contribution of HASARD compared to existing safety benchmarks, such as Safety Gymnasium? Compared to Safety Gymnasium, HASARD primarily adds hard constraints and fast simulation. However, implementing hard constraints is relatively straightforward, merely requiring a single line of code to terminate the episode upon any unsafe action. As for fast simulation, HASARD achieves this by sacrificing simulation fidelity and simplifying the action space, which limits its meaningfulness as a contribution compared to Safety Gymnasium.

Moreover, most tasks in HASARD revolve around avoiding hazardous obstacles, which has already been extensively addressed and solved in Safety Gymnasium by existing algorithms (e.g., [1-3]). Given HASARD's simplified dynamics and action space, it would need to introduce more complex tasks than those in Safety Gymnasium to stimulate the development of new algorithms. However, I did not observe any such complexity in the task design that would distinguish it from prior benchmarks.

[1] CONSTRAINED POLICY OPTIMIZATION VIA BAYESIAN WORLD MODELS

[2] Safe Reinforcement Learning From Pixels Using a Stochastic Latent Representation

[3] SafeDreamer: Safe Reinforcement Learning with World Models

**Questions:**

1. Which tasks in HASARD require memory capabilities, and which involve long-horizon decision-making? It would be helpful if the authors could clarify how the benchmark challenges an agent’s memory and planning capabilities over extended time sequences.

2.  Why did you choose ViZDoom to build this benchmark? Does this platform offer specific advantages? From my perspective, it seems that ViZDoom allows only minor modifications to its existing game structure and may lack the flexibility to define more complex, varied tasks. Why not consider using a truly open-world environment, such as MineDojo [4], which enables safer RL environments with more sophisticated task definitions? A platform like MineDojo could potentially support a broader range of scenarios and facilitate more diverse task creation.

3. Additionally, I noticed that you used Omnisafe for algorithm benchmarking, but this wasn’t mentioned in the paper. I have some questions regarding one of the baselines you implemented. In the P3O algorithm code (see here:https://github.com/PKU-Alignment/omnisafe/blob/main/omnisafe/algorithms/on_policy/penalty_function/p3o.py#L82), there is a term  J_c  in the loss function that appears to be independent of the network parameters. What effect does including J_c in the loss function have? I observed in your experimental results that P3O also fails to satisfy the constraints, which may be related to the J_c term. This raises some doubts about the effectiveness of this baseline.

[4] MineDojo: Building Open-Ended Embodied Agents with Internet-Scale Knowledge

---

> ### Author Response · Authors · 2024-11-24
>
> **W2. Solvability by Existing Algorithms**
>
> **Are the tasks solved?**
> The reviewer has rightfully pointed out the lack of a performance upper bound, without which we cannot determine whether the tasks are already solved. Regular PPO that neglects costs could set a reasonable upper bound for reward, but we are really interested in the highest reward obtainable while adhering to the safety constraints. In our attempt to address this issue, we have included a **human baseline**, similar to the Atari benchmark. A human player was evaluated over 10 episodes on Level 3 of each task, recording both reward and cost. In the following table, we compare the human baseline to the Level 3 main results in the paper obtained with the simplified action space. As noted in Appendix B.1, this had more favorable outcomes than with agents attempting to leverage the full action space. The human player, however, had access to the more complicated original full action space.
>
> |Method|AB R↑|AB C↓|VV R↑|VV C↓|RR R↑|RR C↓|CD R↑|CD C↓|PP R↑|PP C↓|DD R↑|DD C↓|
> |-----------|-------|-------|-------|-------|-------|-------|-------|-------|-------|-------|-------|-------|
> |PPO|1.99|118.22|42.20|347.77|53.16|68.27|34.86|84.44|487.17|894.22|49.36|23.72|
> |PPOCost|*0.04*|*0.05*|*10.61*|*14.76*|*0.01*|*0.02*|*3.93*|*1.35*|*15.39*|*6.64*|49.95|19.79|
> |PPOLag|2.03|*31.89*|22.77|52.54|8.02|9.74|12.22|5.29|23.87|*34.98*|*19.27*|5.26|
> |PPOSauté|2.32|*37.68*|18.89|246.80|*1.17*|*3.50*|*2.74*|*2.76*|101.64|176.14|22.20|10.36|
> |PPOPID|*2.78*|*33.89*|*25.80*|*49.02*|13.51|5.14|*14.51*|*4.97*|*54.02*|*49.57*|*20.49*|*4.87*|
> |P3O|*2.61*|*29.94*|*24.93*|*49.84*|*14.29*|*4.19*|13.57|*4.12*|269.14|428.72|26.73|7.49|
> |Human|***8.03***|*46.43*|***38.06***|*41.12*|***34.21***|*3.17*|***19.43***|*4.77*|***147.32***|*47.45*|***43.90***|*4.02*|
>
> The results indicate that none of the safe RL baselines have achieved human-level performance, suggesting that the tasks are still far from being solved. It is worth noting that the human baseline is suboptimal, leaving ample room for improvement. The sole purpose of this baseline is to show the potential for improved performance, considering also that the human player has privileged knowledge of how the game works. The RL agent has to learn this from pixel observations.
>
> **Why are baselines incapable?**
> In many instances, the human player reached a higher score not due to more precise control, but through strategies that none of the evaluated baselines managed to learn. After all, while the RL method can consistently output precise actions for each frame, human outputs are inherently noisier.
>
> In **Armament Burden**, the agent has the option to discard all weapons. This strategy can be used to get rid of the very heavy weapons far from the starting zone, since carrying them back can be very costly. None of our evaluated baselines managed to learn this behavior. Instead, they simply avoided picking up heavy weapons altogether, as discarding weapons does not provide an immediate or obvious reward. Level 3 introduces decoy items that add to the carrying load but do not grant any reward when delivered. None of the methods were able to figure out how to avoid these items.
>
> In **Precipice Plunge**, the agent has the option to restart the episode, if it decides that no safe actions are possible. This becomes very relevant in Level 3, as the height differences to all surrounding blocks might be too great to permit a safe leap. However, again none of the agents learned to utilize this. Instead, they either remained stuck on a block until the end of the episode or ended up making an unsafe action.
>
> In Levels 2 and 3 of **Remedy Rush**, the environment becomes periodically dark after short time intervals, limiting the agent's ability to see the items it needs to collect or avoid. However, night vision goggles are located in the map, granting permanent vision when obtained. Despite this, none of the baselines developed a consistent strategy to seek out the goggles early in the episode before proceeding with their item collection.
>
> Level 3 of **Detonator's Dilemma** allows the agent to push the barrels to safer locations with few or no units nearby before detonating them. However, since pushing barrels does not yield immediate rewards, the baseline methods failed to adopt this strategy. Instead, the agents tend to avoid going near the barrels, as detonating them while standing too close often results in the agent harming itself and incurring a cost.

---

> ### Author Response · Authors · 2024-11-24
>
> **W3. Task Complexity**
>
> **Focus**
> While we acknowledge that environment suites like Safety-Gymnasium already offer embodied perception environments with more sophisticated physics simulation, our aim is to expand the scope beyond basic navigation and continuous robotic motor control tasks. Rather than competing directly with suites like Safety-Gymnasium, HASARD serves as a complementary benchmark. While safe behavior in realistic physics and lifelike environments is undoubtedly an important direction in the field, HASARD emphasizes fostering higher-order comprehension and decision-making.
>
> The reviewer has raised an important point about the sacrifice of realism - indeed there is a trade-off between the computational efficiency and simulation complexity. HASARD caters for complexity not in terms of the already widely-explored complex continuous control problems, but the ability to perceive, reason, and plan. all the while being magnitudes faster than the complex physics-based simulation environments such as Safety-Gymnasium. This allows researchers to explore other aspects of safe RL at a fraction of the computational cost, especially with the sample-hungry model-free approaches.
>
> **Action Space**
> Regarding the action space, we've introduced 2 settings. We conducted our main experiments using the simplified version of the action space (ranging up to 54 multi-discrete actions, depending on the task). This was done to demonstrate the (1) solvability of the tasks, (2) various reward-cost trade-offs, (3) learning stage analysis, and (4) complexity across difficulty levels. However, the core setting of HASARD is meant to incorporate the full action space of DOOM, featuring **864** multi-discrete actions combined with 2 continuous actions. This significantly raises the difficulty even in Level 1, as we showed in Appendix B.1.
>
> It is arguable whether the continuous action spaces for Safety-Gymnasium agents are more challenging. The Point, Car, and Racecar agents operate with 2 continuous actions (steering and acceleration), Ant uses 8 continuous joint torques, and Doggo has 12 continuous joint torques. Even if this were the case, HASARD does not directly aim to compete with such settings, instead targeting other domains than continuous motor control. The simplified action setting in HASARD is meant to streamline development by enabling faster analysis, quicker iteration, and immediate feedback. It allows researchers to test and refine methods efficiently, focusing on task dynamics without the computational overhead of the full-action space.
>
> **Hard Constraints**
> We appreciate the reviewer for addressing this. We wish to clarify that introducing hard constraints is not a novel contribution of our work, nor is it a central focus of this paper, as similar setups can be implemented in other libraries with little ease. Do note, however, that our modifications vary between tasks and go beyond only lowering the cost threshold to zero. For example, in *Armament Burden*, exceeding the carrying capacity results in the loss of all previously collected weapons. The aim of the hard constraint setting is to provide a wider variety of evaluation settings, due to various safety formulations. We will reduce the emphasis and claims regarding the hard constraints in the writing.
>
> The responses to the following weaknesses and questions will further describe the challenges of the HASARD scenarios.

---

> ### Author Response · Authors · 2024-11-24
>
> **W4. Only Navigation Tasks**
>
> Indeed, most tasks in popular benchmarks such as Safety-Gymnasium (1st-person POV) [1], BulletSafetyGym (proprioceptive) [2], and MetaDrive (3rd-person POV) [3] primarily focus on the agent navigating toward a goal location while avoiding hazardous objects. However, they are comparatively simpler to solve. For example, many Safe RL methods that rely on a cost critic, only require it to learn to assign low values to observations where hazards are prominently visible in the center of the screen, and higher values to observations where hazards are either absent or located at the edges. This allows the critic to provide accurate value estimates based on a single observation, without requiring any history or further context.
>
> In contrast, the scenarios in HASARD require a deeper level of perception and decision-making. Agents must comprehend spatial relationships, accurately perceive depth, and predict the movement and future locations of entities. Only the **Remedy Rush** task has a navigation objective similar to the prior benchmarks. However, it features more complex visual surroundings, a greater variety of items to collect, and objects to avoid. In contrast to Safety-Gymnasium’s visually simplistic environments, with distinct, single-palette cylindrical goals and hazard zones that are easy to differentiate, the collectibles and hazardous objects in *Remedy Rush* are more difficult to identify.
>
> The other 5 tasks each have their unique nature and distinct objective. **Volcanic Venture** could be considered most similar, as the agent does have to avoid certain zones when navigating the area. However, the platforms, the agent should stay on, periodically switch locations, are of different height and in constant motion. These attributes far extend the static Safety-Gymnasium environments.
>
> **Armament Burden** introduces another layer of partial observability on top of the conventional limited field of view with egocentric perception. The agent has a carrying load that is not in any way observable from the pixels on the screen. For instance, acquiring a weapon with an empty inventory incurs no immediate cost. However, if the agent has already picked up other items, this could exceed its carrying capacity.
>
> Similarly, in **Detonator's Dilemma**, detonating a barrel near a neutral unit with high health points might result in minimal consequences, whereas doing so near units with low HP could incur high cost. The agent should not only accurately perceive the distance between the neutral units and the barrels, but also the barrels themselves, to prevent unintended chain explosions.
>
> The core difficulty in **Collateral Damage** lies in accurately firing the weapon, as the projectile takes time to reach its destination. Both neutral and enemy units are in constant motion, making it difficult to predict their future positions in relation to the projectile. The agent must also balance risks and rewards, considering whether targeting a cluster of enemies near neutral units is worth the cost within the allowable budget.
>
> In **Precipice Plunge**, the central challenge lies in depth perception, as the agent must find a safe path to carefully descend down the cave and avoid fall damage. This type of task is explored in any of the prior benchmarks. As the difficulty level increases, the randomized terrain becomes steeper and some blocks are in constant vertical movement. Moreover, the cave becomes progressively darker as the agent ventures deeper.
>
> The answer to Q1 will further outline the challenges of HASARD environments. Moreover, in Appendix A of the paper, we have described the characteristics and complexities of each environment.
>
> [1] Ji, Jiaming, et al. "Safety gymnasium: A unified safe reinforcement learning benchmark." Advances in Neural Information Processing Systems 36 (2023).
>
> [2] Gronauer, Sven. "Bullet-safety-gym: A framework for constrained reinforcement learning." (2022).
>
> [3] Li, Quanyi, et al. "Metadrive: Composing diverse driving scenarios for generalizable reinforcement learning." IEEE transactions on pattern analysis and machine intelligence 45.3 (2022): 3461-3475.

---

> ### Author Response · Authors · 2024-11-24
>
> **Q1. Which Tasks Require Memory and Planning?**
>
> In **Armament Burden**, the agent needs to memorize which items it has already collected to estimate how many more, and of what type, it is reasonable to gather before returning to the delivery zone. There is no direct indicator of the carried weight from a raw observation. The only feedback is the slowdown the agent experiences once the capacity is exceeded. When setting off from the starting zone, the agent should plan out a route of which items in which order to obtain to maximize its reward in a limited time. The weapon types and their locations are random each time, presenting a unique setting at every start. Due to partial observability, especially in higher levels, where the entire environment and obtainable items are not visible from the starting zone, it is advantageous for the agent to memorize the locations of items it has seen but not obtained in preceding runs during the same episode.
>
> In **Collateral Damage**, the movement and velocity of enemy and neutral units can not be inferred from a single frame, necessitating a capacity for short-term memory. Each entity type exhibits different movement patterns that can be anticipated to some extent. By accurately perceiving depth and predicting future positions of the entities, the agent can plan its actions and time its rocket launches effectively to hit targets while avoiding collateral damage to neutral units.
>
> Similar to predicting the sporadic movements of units in *Collateral Damage*, the agent must anticipate unit trajectories in **Detonator's Dilemma**, accompanied by a greater variety in unit behaviors and a more complex terrain. With barrels scattered across the map and both neutral units and barrels respawning after elimination, the agent benefits from maintaining a memory of prior detonations and unit locations to optimize its strategy. For instance, if the agent observes that its immediate area has numerous barrels but is heavily crowded with neutral units, it might strategically navigate to a less populated area to detonate barrels there. By retaining this memory, the agent can later return to the original location when conditions are more favorable.
>
> We will further expand the environment descriptions in Appendix A to incorporate these details. If the reviewer feels that our claims regarding memory and planning capabilities are overstated or artificial, we are open to revising the emphasis.
>
>
> **W1. Outdated Baselines**
>
> We appreciate the reviewer’s concern regarding the use of recent baselines. The baseline methods we selected were chosen for their robustness and proven track record across a variety of prior benchmarks. While these methods may not represent the absolute latest developments, they are well-tested and reliable for evaluating the complexity and nuances of HASARD. That said, we acknowledge that integrating more recent vision-based safe RL methods could enhance the relevance of HASARD as a benchmark.
>
> While Lambda, Safe SLAC, and SafeDreamer have public author-provided implementations, they are not yet integrated into widely-used Safe RL libraries such as SSA [1], FSRL [2], SafePO [3], and OmniSafe [4], which could streamline their adoption. Adapting them to new environments like HASARD requires substantial effort in terms of implementation adjustments.
>
> Additionally, we encountered compatibility issues during our attempts to incorporate these methods. For instance, SafeDreamer relies on JAX [5], which is well-optimized for environments natively written in JAX but presents challenges when integrating with non-native environments. Similarly, neither Lambda nor Safe SLAC could be installed directly using the dependencies listed in their respective repositories. Despite extensive efforts and manual adjustments, we were unable to get these methods to train effectively.
>
> [1] Ray, Alex, Joshua Achiam, and Dario Amodei. "Benchmarking safe exploration in deep reinforcement learning." arXiv preprint arXiv:1910.01708 7.1 (2019): 2.
>
> [2] Liu, Zuxin, et al. "Datasets and benchmarks for offline safe reinforcement learning." arXiv preprint arXiv:2306.09303 (2023).
>
> [3] https://github.com/PKU-Alignment/Safe-Policy-Optimization
>
> [4] Ji, Jiaming, et al. "Omnisafe: An infrastructure for accelerating safe reinforcement learning research." Journal of Machine Learning Research 25.285 (2024): 1-6.
>
> [5] https://github.com/PKU-Alignment/SafeDreamer

---

> ### Author Response · Authors · 2024-11-24
>
> **Q2. Why ViZDoom?**
>
> The core advantages of ViZDoom lie in its speed, lightweight design, cross-platform compatibility, and ease of configuration. Although it may lack realism, it effectively replicates the notion of navigation and interaction in a 3D realm, which is directly tied to real-world problems. ViZDoom remains a widely adopted platform in the RL community and continues to be used [1, 2, 3, 4]. Indeed, as the reviewer has pointed out, it is useful for the platform to be highly customizable for the creation of more diverse tasks. In Appendix E, we have listed several promising extensions for HASARD. Apart from increasing task complexity through configurable parameters, one could combine challenging elements across tasks. The rationale behind selecting ViZDoom as the foundation for HASARD is further elaborated in Appendix F.
>
> **Open-World**
> In the context of the tasks we designed, the open-world aspect does not significantly impact the Safe RL challenges. Whether the engine procedurally generates more content as the agent ventures further or whether the agent eventually hits a wall has only a marginal effect on task complexity. As demonstrated and discussed earlier, the higher levels of HASARD or the full action space setting already provide a sufficient challenge to current methods. Conversely, having a physically bounded environment enables a more detailed analysis of the agent's movement patterns, allowing us to draw connections between its behavior and the training process or the specific method employed, as examined in Section 5.4. Moreover, tasks with a more bounded scope enhance reproducibility and provide more fair ground for comparison.
>
> Open-world environments like MineDojo are computationally more expensive to run, making them less suitable for limited resources. ViZDoom, on the other hand, uses rendering with sprites and precomputed lighting, making it computationally inexpensive, lightweight, and accessible, while open-world environments often use dynamic lighting, shadows, and voxel-based rendering, adding a computational overhead.
>
>
> **MineDojo**
> We agree with the reviewer that a Minecraft-based benchmark holds significant potential for evaluating Safe RL and could challenge similar competencies that HASARD aims to address. Minecraft offers great variety in terms of items, entities, interactions, and possible objectives. A key advantage of MineDojo lies in its extensive knowledge base, built from gameplay videos and the Minecraft Wiki, which could be effectively leveraged for offline Safe RL methods.
>
> However, for online learning, the simulation speed might pose challenges. Although neither DOOM nor Minecraft were created for RL experiments, Vanilla Minecraft has notable downsides for RL. Its Java-based engine introduces layers of abstraction, memory management overhead, and rendering complexity, which negatively impact performance. Additionally, while ViZDoom allows efficient parallel execution of multiple instances, Minecraft’s Java implementation with its higher memory usage and single-threaded limitations—makes scaling less efficient. Nonetheless, we recognize a Minecraft-based Safe RL benchmark as a promising avenue for future work.
>
> **Q3. How is P3O Implemented?**
>
> We apologize for any confusion. To clarify, we used the Sample-Factory [5] framework for algorithm benchmarking, not Omnisafe. As for P3O, whether it is generally an effective method due to its algorithmic details falls outside the scope of our work. Our focus is on evaluating widely known popular algorithms within our proposed benchmark rather than assessing the intrinsic effectiveness of individual algorithms like P3O.
>
> [1] Park, Junseok, et al. "Unveiling the Significance of Toddler-Inspired Reward Transition in Goal-Oriented Reinforcement Learning." Proceedings of the AAAI Conference on Artificial Intelligence. Vol. 38. No. 1. 2024.
>
> [2] Kim, Seung Wook, et al. "Neuralfield-ldm: Scene generation with hierarchical latent diffusion models." Proceedings of the IEEE/CVF conference on computer vision and pattern recognition. 2023.
>
> [3] Zhai, Yunpeng, et al. "Stabilizing Visual Reinforcement Learning via Asymmetric Interactive Cooperation." Proceedings of the IEEE/CVF International Conference on Computer Vision. 2023.
>
> [4] Valevski, Dani, et al. "Diffusion models are real-time game engines." arXiv preprint arXiv:2408.14837 (2024).
>
> [5] Petrenko, Aleksei, et al. "Sample factory: Egocentric 3d control from pixels at 100000 fps with asynchronous reinforcement learning." International Conference on Machine Learning. PMLR, 2020.

---

> ### Comment · Reviewer_FzXY · 2024-11-26
>
> Thanks for the reply. Considering that the contribution of this paper is incremental, I decided to increase my score. However, the author did not address my biggest concern and try to use the latest algorithm to run the proposed benchmark environment during the rebuttal period, so it's still unclear whether the current algorithm has solved the problem proposed by this benchmark.

---

> > ### Author Response · Authors · 2024-12-04
> >
> > Thank you for reviewing our response. To address your biggest concern, we have included **SafeDreamer** [1], the most recent state-of-the-art Safe RL method, in our evaluation. Similarly to other benchmarks, SafeDreamer surpasses prior baselines, achieving the highest results across all environments. However, it only surpasses the human baseline on *Collateral Damage*, a task requiring high precision and timing. It does yet not manage to develop clever strategies in *Armament Burden* (discarding heavy weapons and avoiding decoy items), *Precipice Plunge* (restarting the episode when a safe action can no longer be performed), and *Remedy Rush* (immediately seeking out the night vision goggles to gain permanent vision). Therefore, in these environments the human baseline, although weaker in control and accuracy, still outperforms SafeDreamer thanks to a better strategy and knowledge of the task. The results of evaluating SafeDreamer show, that HASARD environments provide ample room for algorithmic improvement, not to mention when also utilizing the full action space with 864 discrete and 2 continuous actions, which could make control far more efficient. We hope that the inclusion of SafeDreamer is sufficient to satisfy the reviewer's final concern.
> >
> > |Method|AB R↑|AB C↓|VV R↑|VV C↓|RR R↑|RR C↓|CD R↑|CD C↓|PP R↑|PP C↓|DD R↑|DD C↓|
> > |-----------|-------|-------|-------|-------|-------|-------|-------|-------|-------|-------|-------|-------|
> > |PPO|1.99|118.22|42.20|347.77|53.16|68.27|34.86|84.44|487.17|894.22|49.36|23.72|
> > |PPOCost|*0.04*|*0.05*|*10.61*|*14.76*|*0.01*|*0.02*|*3.93*|*1.35*|*15.39*|*6.64*|49.95|19.79|
> > |PPOLag|2.03|*31.89*|22.77|52.54|8.02|9.74|12.22|5.29|23.87|*34.98*|*19.27*|5.26|
> > |PPOSauté|2.32|*37.68*|18.89|246.80|*1.17*|*3.50*|*2.74*|*2.76*|101.64|176.14|22.20|10.36|
> > |PPOPID|*2.78*|*33.89*|*25.80*|*49.02*|13.51|5.14|*14.51*|*4.97*|*54.02*|*49.57*|*20.49*|*4.87*|
> > |P3O|*2.61*|*29.94*|*24.93*|*49.84*|*14.29*|*4.19*|13.57|*4.12*|269.14|428.72|26.73|7.49|
> > |**SafeDreamer**|*3.11*|*48.48*|*33.07*|*49.60*|*19.81*|*4.79*|***21.57***|*4.77*|71.22|44.03|36.33|*4.60*|
> > |Human|***8.03***|*46.43*|***38.06***|*41.12*|***34.21***|*3.17*|19.43|*4.77*|***147.32***|*47.45*|***43.90***|*4.02*|
> >
> > [1] Huang, Weidong, et al. "Safe dreamerv3: Safe reinforcement learning with world models." arXiv preprint arXiv:2307.07176 (2023).

---

### Meta-Review · Area_Chair_CS2t · 2024-12-20

**Metareview:**

This paper presents a new benchmark intended to be used for evaluating safe reinforcement learning algorithms, built on the ViZDoom platform. The benchmark introduces a variety of new environments and tasks. Experiments with a few baselines illustrate the headroom on the benchmark (current methods do not saturate it, but also achieve non-trivial performance). The claimed contribution is that it is the first to exclusively target vision-based safe RL.

### Strengths
Reviewers commented on the variety of environments and challenges introduced and the utility of the structured curriculum (three difficulty levels), analysis of a variety of baselines and that the benchmark requires solving tasks beyond simple 2D navigation, that the paper targets the important / significant problem of building realistic and reliable RL and safe RL benchmarks, detailed evaluations of a variety of baselines, the originality of the contribution, and the comprehensive design

### Weaknesses
Reviewers commented that some of the baselines were outdated, potential solvability by existing algorithms, unclear contribution relative to the Safety Gymnasium benchmark (including that most of the tasks revolve around avoiding hazardous obstacles), that the baselines were narrow, there was limited visual input analysis required, the action space was limited, and that the connection between the benchmark and real-world tasks was unclear.

The authors responded to these weaknesses (see my summary of the details below), and in balance, sufficiently addressed and dispelled arguments for rejection.

**Additional Comments On Reviewer Discussion:**

Reviewer FzXY (rating 8, confidence 4) found their concerns addressed by the authors

Reviewer NMGZ (rating 5, confidence 4) increased their score from 3 to 5 after the rebuttal. The reviewer did not clarify what concerns remained when prompted by the authors.

Reviewer Rd3K (rating 5, confidence 4) remained unconvinced by the significance of the benchmark, stating
> I am not convinced the safe RL community will be incentivized to use it. The main reason is that the notions of constraints in this benchmark are not directly tied to the very pragmatic safety considerations that need to be tackled in the real world - ranging from control systems to robotic deployments.

and
> The benchmark feels a bit incremental compared to the already existing VizDoom framework that has been around for years.

and
> I am still not convinced about the significance of the proposed benchmark, the applicability of algorithms tested on it to other more realistic scenarios where safety would be actually meaningful

and

> I do not see how any form of safety guarantees from this environment can realistically translate to these applications that the authors have cited

The crux of these concerns seems to stem from the lack of physical realism (no simulated physics in the benchmark). However, while the reviewer may turn out to be correct that the safe RL community may not use the benchmark, I do not think this speculation is sufficient to warrant rejection. Because the paper and benchmark are otherwise solid and have clearly new contributions, I would err on the side of letting the safe RL community decide whether it is a meaningful benchmark. Furthermore, regarding the "form of safety guarantees" comment from the reviewer, the authors did not make the claim that the benchmark's value would be to develop methods that provided safety guarantees (I do not think that that is an agreed-upon definition of the purpose of safe RL research / methods). It is plausible that this benchmark could serve as an important and much-needed vehicle for future safe RL research that produces safe RL methods that contain no safety guarantees (e.g., those that only deliver empirically high performance).

Reviewer NtcD (rating 5, confidence 3) did not respond to the author's response. The authors, in my judgement, convincingly addressed their concerns.

---

### Decision · Program_Chairs · 2025-01-22

Accept (Poster)